# GAS: Enhancing Reward-Cost Balance of Generative Model-assisted Offline Safe RL

**Zifan Liu**[1]**, Xinran Li**[1]**, Shibo Chen**[2*]**, Jun Zhang**[1*]
[1] The Hong Kong University of Science and Technology, Hong Kong SAR, China
[2] South China University of Technology, Guangdong, China

## Abstract

Offline Safe Reinforcement Learning (OSRL) aims to learn a policy that achieves high performance in sequential decision-making while satisfying safety constraints, using only pre-collected datasets. Recent works, inspired by the strong capabilities of Generative Models (GMs), reformulate decision-making in OSRL as a conditional generative process, where GMs generate desirable actions conditioned on predefined reward and cost return-to-go values. However, GM-assisted methods face two major challenges in constrained settings: (1) they lack the ability to "stitch" optimal transitions from suboptimal trajectories within the dataset, and (2) they struggle to balance reward maximization with constraint satisfaction, particularly when tested with imbalanced human-specified reward-cost conditions. To address these issues, we propose Goal-Assisted Stitching (GAS), a novel algorithm designed to enhance stitching capabilities while effectively balancing reward maximization and constraint satisfaction. To enhance the stitching ability, GAS first augments and relabels the dataset at the transition level, enabling the construction of high-quality trajectories from suboptimal ones. GAS also introduces novel goal functions, which estimate the optimal achievable reward and cost goals from the dataset. These goal functions, trained using expectile regression on the relabeled and augmented dataset, allow GAS to accommodate a broader range of reward-cost return pairs and achieve a better tradeoff between reward maximization and constraint satisfaction compared to human-specified values. The estimated goals then guide policy training, ensuring robust performance under constrained settings. Furthermore, to improve training stability and efficiency, we reshape the dataset to achieve a more uniform reward-cost return distribution. Empirical results validate the effectiveness of GAS, demonstrating superior performance in balancing reward maximization and constraint satisfaction compared to existing methods.

## 1 Introduction

Significant progress has been achieved in Reinforcement Learning (RL) that learns policies to maximize rewards through constant interactions with the environment. Limited by the trial-and-error approach, standard RL often fails in scenarios with safety constraints, such as autonomous driving (Fang et al., 2022) and investment portfolios (Lee & Moon, 2023). It stems from two issues: (1) the exploration process in RL can be inherently risky, as random actions may lead to unsafe outcomes, and (2) the optimized policies often focus solely on maximizing cumulative rewards, neglecting safety constraints. To tackle these issues, Offline Safe RL (OSRL) has emerged as a promising paradigm that learns safe policies from a pre-collected dataset, eliminating the need for risky online exploration.

While OSRL mitigates the risks associated with online exploration by relying on static datasets, it introduces a new challenge: the Out-Of-Distribution (OOD) problem. Specifically, the Bellman backup may extrapolate actions beyond the dataset, leading to unpredictable or unsafe behaviors. Early OSRL methods primarily adapted RL techniques to constrained settings using approaches such as the Lagrange method (Stooke et al., 2020), constraint penalty (Xu et al., 2022), or the DICE-style techniques (Lee et al., 2022). To mitigate OOD-related issues, these methods incorporate strategies like distribution correction (Lee et al., 2022), regularization (Kostrikov et al., 2021), and OOD detection (Xu et al., 2022). However, they still face challenges in effectively addressing OOD problems and adapting to dynamic, real-world constraints.

---

*Both Shibo Chen (shibochen.ustc@gmail.com) and Jun Zhang (eejzhang@ust.hk) are corresponding authors.

More recently, Generative Model-assisted (GM) methods have emerged as an alternative to traditional OSRL approaches to address these limitations. In OSRL, GM methods reformulate the Constrained Markov Decision Process (CMDP) as a goal-conditioned generating problem, where the generative model is trained to produce trajectories that match predefined reward and cost returns specified as inputs. This formulation provides two key benefits. First, GM methods essentially adopt a goal-conditioned behavior cloning scheme, where the model learns to imitate behavior from the dataset conditioned on the desired reward and cost targets. This formulation entirely bypasses the Bellman backup procedure, which is the primary source of the OOD problem in traditional OSRL approaches. Second, GM methods also offer greater flexibility compared to conventional methods, which typically operate under fixed safety constraints. Because the target reward and cost returns are provided as inputs, GM methods can seamlessly adapt to varying objectives at test time without re-training.

Nevertheless, GM methods present their own challenges. First, while GM methods bypass the Bellman backup procedure, they lack trajectory stitching capabilities (Badrinath et al., 2023; Wu et al., 2023; Kim et al., 2024b) — the ability to combine transitions from different trajectories to enhance performance. This limitation restricts their ability to fully utilize suboptimal datasets to improve performance. Second, GM methods lack an explicit mechanism to balance reward maximization and constraint satisfaction, potentially leading to unsafe or overly conservative policies. In this work, we propose a novel *Goal-Assisted Stitching (GAS)* method to address these challenges while retaining the advantages of GM methods. The key contributions are summarized as follows.

We propose to use goal functions as intermediate values to bridge the gap between human-specified (potentially suboptimal) reward-cost targets and the optimal achievable goals of the conditional policy instantiated by GM methods. To achieve this, we introduce three key innovations: **1) Temporal Segmented Return Augmentation and Transition-level Return Relabeling:** To enhance GAS's stitching capabilities, we restructure the offline dataset at the transition level and introduce temporal segmented return augmentation, which extracts richer information by considering reward and cost returns over varying timesteps rather than only at trajectory endpoints. Additionally, we ensure robustness to suboptimal human-specified target return-to-goes during testing by relabeling reward-cost return-to-goes at the transition level during training. **2) Goal Functions with Expectile Regressions:** We train the novel reward and cost goal functions using expectile regression to estimate the optimal achievable reward and cost goals without relying on Bellman backups. These goal functions guide the policy to stitch transitions effectively, achieving both reward maximization and constraint satisfaction for a wider range of given target reward-cost return pairs. **3) Dataset Reshaping:** To improve training stability and efficiency, we address the data imbalance issue by reshaping the dataset to create a more balanced reward-cost return distribution. **Through extensive experiments on 2 benchmarks with 12 scenarios and 8 baselines under various constraint thresholds, GAS shows superior safety ability under tight thresholds and 6% improvement in performance under loose thresholds.**

## 2 RELATED WORK

**RL Methods in Offline RL.** Offline RL (Fujimoto et al., 2019) aims to find the policy to maximize the cumulative rewards from a pre-collected dataset. The primary challenge lies in the OOD problem during the Bellman backup, where the policy may select actions beyond the dataset. This issue arises because offline RL does not allow for environment exploration to gather additional data. To this end, most existing works try to constrain the target policy to stay close to the behavior policy with a KL regularization term (Jaques et al., 2020; Peng et al., 2019; Siegel et al., 2020) or Wasserstein distance (Wu et al., 2020) where the behavior policy is estimated by a generative model. For example, BCQ (Fujimoto et al., 2019) uses a variational autoencoder (VAE) to estimate the behavior policy and restrict the action space during the Bellman backup. Except for explicitly estimating the behavior policy, CQL (Kumar et al., 2020) proposes to learn Q-values conservatively, encouraging those within the dataset to act as a lower bound. Alternatively, IQL (Kostrikov et al., 2022) avoids the OOD problem entirely by employing expectile regression to learn Q-values without querying OOD actions.

**GM Methods in Offline RL.** More recently, some works formulate offline RL as a return-conditioned generation problem and address it using generative models, such as Decision Transformer (DT) (Wu et al., 2023; Chen et al., 2021; Zheng et al., 2022), and Decision Convformer (DC) (Kim et al., 2024a). These methods naturally avoid the OOD problem since the return-conditioned generation problem does not need the Bellman backup procedure. However, a significant limitation of these methods is their limited stitching ability, which refers to the ability to find better policies beyond the dataset trajectories, particularly when the dataset consists primarily of suboptimal trajectories. This definition

is a little different from the stitching ability in RL since GM methods try to avoid the Bellman backup procedure. To improve the stitching ability of GM methods, several works have been proposed. For instance, QDT (Yamagata et al., 2023a) relabels return-to-go in the dataset with Q-values derived from RL methods. WT (Badrinath et al., 2023) proposes to use a sub-goal as the prompt to guide the DT policy to find a shorter path in navigation problems. Building on these ideas, ADT (Ma et al., 2024) proposes a hierarchical framework by replacing the return-to-go with a sub-goal learned from IQL. Reinformer (Zhuang et al., 2024) generates RTG based on the state using expectile regression and achieves strong reward stitching capabilities. In contrast to improving stitching ability in the training procedure, EDT (Wu et al., 2023) boosts the stitching ability of DT at decision time by adaptively adjusting the history length of the attention module.

**Offline Safe RL.** OSRL integrates safety constraints into offline learning settings, addressing both safety requirements and limited online interaction with the environment. This emerging field has spawned two main approaches: RL methods and GM methods. For RL methods, CPQ (Xu et al., 2022) employs a conditional VAE model to estimate and penalize the OOD actions. COptiDICE (Lee et al., 2022) utilizes stationary distribution correction to mitigate the distributional shift problem. For GM methods, which are inspired by DT, CDT (Liu et al., 2023b) transforms OSRL into a goal-conditioned generative problem and inputs both target reward and cost return-to-go to the GPT structure. To handle the safety-critical cases, FISOR (Zheng et al., 2024) proposes a feasibility-guided diffusion model to ensure strict satisfaction of constraints.

Although previous works demonstrate great success in enhancing the stitching ability of GM methods in offline RL, they cannot be utilized in OSRL directly due to the fundamentally different objectives introduced by constrained settings. Extending GM methods to OSRL is challenging because, while the goal of maximizing rewards remains consistent, the requirements for constraints vary across scenarios. CQDT (Wang & Zhou, 2025) tries to improve the CDT by training lots of RL policies under each constraint, which is extremely inefficient. Recently, a concurrent work, named COPDT (Xue et al., 2025) is proposed by considering the reward targets as a generation conditioned on cost targets to prioritize cost targets, focusing on multi-constraints and multi-tasks. Different from these works, we address this critical gap by introducing GAS, a novel framework designed to give up attention structure and purely focus on "transition stitching" only with a simple Multi-Layer Perceptron (MLP) structure guided by robust reward and cost goal functions.

## 3 BACKGROUND

**Constrained Markov Decision Process (CMDP).** CMDP (Altman, 1998) is a standard framework for safe RL defined by the tuple $\mathcal{M} = (\mathcal{S}, \mathcal{A}, \mathcal{P}, r, c, \gamma)$, where $\mathcal{S}$ is the state space, $\mathcal{A}$ is the action space, $\mathcal{P}(s'|s, a)$ is the transition function, $r(s, a)$ is the reward function, $c(s, a)$ is the cost function, and $\gamma$ is the discount factor. The goal of safe RL is to find a policy $\pi(a|s)$ that maximizes the cumulative rewards while satisfying the safety constraints. Given the constraint threshold $L$ and the trajectory $\tau = \{(s_0, a_0, r_0, c_0), ..., (s_T, a_T, r_T, c_T)|a_t = \pi(a_t|s_t)\}$, with $T$ being the trajectory length, the optimization problem of safe RL can be written as:

$$\max_{\pi} V_r^{\pi}(\tau),$$
$$s.t. \ V_c^{\pi}(\tau) \leq L, \tag{1}$$

where $V_r^{\pi}(\tau) = \mathbb{E}_{\tau \sim \pi}[\sum_{t=0}^{\infty} \gamma^t r_t]$ denotes the reward value function, and $V_c^{\pi}(\tau) = \mathbb{E}_{\tau \sim \pi}[\sum_{t=0}^{\infty} \gamma^t c_t]$ denotes the cost value function.

**Offline Safe Reinforcement Learning (OSRL).** OSRL maintains the same objectives as safe RL while additionally addressing the OOD problem due to the inability to explore. To mitigate the OOD problem, the target policy is constrained to remain close to the behavior policy, as formulated in eq. (2), which augments eq. (1) in the optimization.

$$\mathbb{D}(\pi, \mu) \leq \zeta, \tag{2}$$

where $\mu$ is the behavior policy used in the pre-collected dataset $\mathcal{D}$, $\mathbb{D}(., .)$ is an arbitrary distance/divergence function, and $\zeta$ is a hyper-parameter.

**Constrained Decision Transformer (CDT).** CDT (Liu et al., 2023b) transfers the OSRL problem into a goal-conditioned generative problem by reformulating the dataset as:

$$\tau = (s_0, a_0, R_0, C_0, ..., s_T, a_T, R_T, C_T), \tag{3}$$

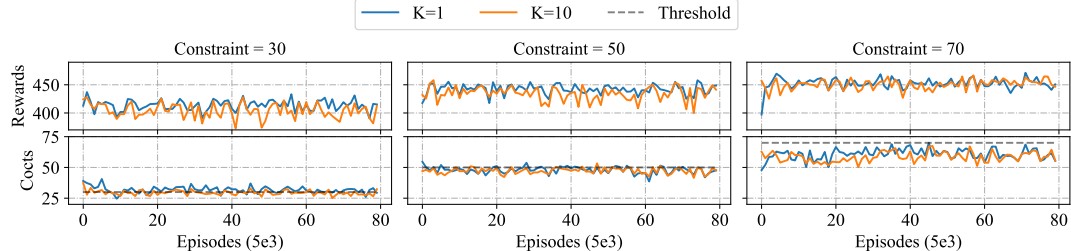

Figure 1: Training curve of CDT with different memory length $K$ on task *CarCircle*.

where $R_t = r_t + ... + r_T$ is the cumulative rewards from $t$ to the trajectory end in the dataset, named reward return-to-go, and $C_t = c_t + ... + c_T$ is the cost return-to-go. Then CDT trains a policy $\pi$ with the GPT structure, as shown in eq. (4), to predict the action given $(s, R, C)$ at the current timestep and $(s, a, R, C)$ from previous $K - 1$ timesteps as conditions, where $K$ is the memory length.

$$\pi(a_t|s_t, R_t, C_t, ..., a_{t-K+1}, s_{t-K+1}, R_{t-K+1}, C_{t-K+1}) \quad \text{a.k.a} \quad \pi(a_t|s_t, R_t, C_t, K), \quad (4)$$

During testing, users need to specify the target reward return $\hat{R}$ and cost return $\hat{C}$ as the input of the CDT policy, which then generates actions to approach these targets $\hat{R}$ and $\hat{C}$.

## 4 MOTIVATION

In this section, we first analyze two critical limitations of current GM methods for OSRL: (1) insufficient trajectory stitching capabilities and (2) the inability to balance reward maximization and constraint satisfaction. We design our solution to specifically address these challenges.

### 4.1 INSUFFICIENT TRAJECTORY STITCHING CAPABILITIES

A key strength of traditional RL methods lies in their ability to stitch together different suboptimal trajectories, enabling effective generalization across diverse experiences. This stitching ability arises from the fact that RL only considers the current state as the condition and stitches the next optimal transition via Bellman backup. In contrast, GM methods sacrifice this crucial ability by additionally taking previous information as conditions to capture the temporal association among transitions. Recent studies (Badrinath et al., 2023; Ma et al., 2024; Xiao et al., 2023; Yamagata et al., 2023b) demonstrate that DT/CDT's trajectory-level training paradigm is essentially a form of goal-conditioned behavior cloning, which predicts the action based on information in previous timesteps via the temporal attention module. Given suboptimal information as the contextual condition, GM methods tend to memorize the suboptimal actions, resulting in suboptimal performance.

However, Kim et al. (2024b) reveals that the attention module designed for natural language processing often fails to characterize the temporal relations for sequential transitions of MDPs in DT (Chen et al., 2021). Specifically, while MDP policies should primarily focus on immediate previous states, DT's attention spans up to 20 timesteps backward, potentially diluting the focus on relevant temporal information and adversely affecting the performance of offline RL (Kim et al., 2024b). To validate this observation, we conducted empirical experiments comparing CDT variants with different memory lengths ($K = 1$ and $K = 10$) for attention modules while keeping other parameters constant. As shown in fig. 1, CDT's performance remains largely unchanged across these memory settings under three distinct constraints, suggesting that the attention mechanism fails to effectively leverage temporal information in OSRL.

### 4.2 INABILITY TO BALANCE REWARD MAXIMIZATION AND CONSTRAINT SATISFACTION

In GM-assisted OSRL, as opposed to goal-conditioned behavior cloning, the objective should be formulated as:

$$\max_{\pi} \mathbb{R}(s_t, \pi(a_t|s_t, R_t, C_t, K))$$
$$s.t. \mathbb{C}(s_t, \pi(a_t|s_t, R_t, C_t, K)) \leq C_t,$$
$$R_t = \hat{R} - (r_0 + ... + r_{t-1}), \quad (5)$$
$$C_t = \hat{C} - (c_0 + ... + c_{t-1}),$$

where $\mathbb{R}(.)$ and $\mathbb{C}(.)$ are the non-discounted reward and cost returns under the policy $\pi$. DT-family algorithms, due to their goal-conditioning nature (detailed in section 3), struggle to effectively balance

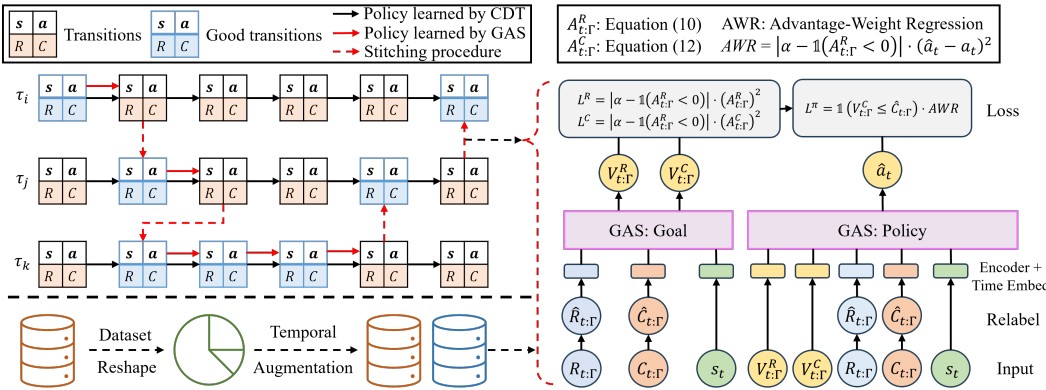

Figure 2: Overall view of GAS. Left: Comparison between CDT and GAS. CDT optimizes at a trajectory level, while GAS enables fine-grained trajectory stitching under the guidance of reward maximization and constraint satisfaction. Right: GAS's stitching mechanism, where the goal function learns the optimal reward and cost return-to-go within the constraint given any target. The policy aims to take actions to achieve optimal goals estimated by the goal function via constrained AWR.

the dual objectives of reward maximization and constraint satisfaction in OSRL. This limitation stems from two key challenges: **First**, without prior knowledge, it becomes problematic to determine appropriate target reward-cost pairs $(\hat{R}, \hat{C})$ as these objectives often require careful trade-offs. The GPT structure can only search for policies that satisfy given targets (Liu et al., 2023b) without validating their feasibility. Consequently, specifying overly ambitious return targets $\hat{R}$ alongside stringent constraints $\hat{C}$ may lead to policy degradation when such conditions prove unrealistic. **Second**, the current CDT architecture simply concatenates target return $\hat{R}$ and cost $\hat{C}$ as contextual conditions without any mechanism to balance their relative importance. This structural limitation prevents GM methods from properly prioritizing constraint satisfaction over reward maximization—a crucial requirement in OSRL. This misalignment between architectural design and OSRL objectives frequently causes performance degradation in practice.

## 5 GOAL-ASSISTED STITCHING

In section 4.1, we highlight that extending the temporal attention module to overly long time windows does not enhance memorization and instead hinders the stitching ability for GM methods in OSRL. This inspires us to trade the temporal attention for a more focused approach: stitching transitions in the dataset under the guidance of reward maximization and constraint satisfaction in a conditional manner. To achieve this goal, we propose Goal-Assisted Stitching (GAS), a novel algorithm that offers a more flexible balance between $R_t$ and $C_t$ and stitches high-quality transitions to achieve safe and best performance as shown in fig. 2. In particular, we first estimate the optimal achievable reward return-to-go satisfying the constraint and corresponding cost return-to-go in the dataset for a given pair of $R_t, C_t$ with a reward-cost goal function through expectile regression without relying on the Bellman backup procedure. The estimated optimal reward and cost goals are then used to guide the policy optimization through a constrained policy optimization paradigm. To further enhance the stability and efficiency of the training procedure, we reshape the dataset in terms of $R_t, C_t$ distribution, ensuring a more uniform reward-cost-return distribution to support GAS training. **Theoretical analysis and the pseudocode of GAS are presented in section B and section C.**

### 5.1 OPTIMAL ACHIEVABLE GOALS

Conceptually, the maximum achievable return-to-go $V_t^R$ and the corresponding cost return-to-go $V_t^C$ for a given state $s$ and the target reward/cost returns $\hat{R}$ and $\hat{C}$ in the dataset should follow:

$$V_t^R(s, \hat{R}, \hat{C}) = \max_{(s_t = s, a_t, R_t, C_t) \sim \mathcal{D}} R_t * \mathbb{1}(C_t \leq \hat{C}). \tag{6}$$

$$V_t^C(s, \hat{R}, \hat{C}) = \arg_{C_t} \{ \max_{(s_t = s, a_t, R_t, C_t) \sim \mathcal{D}} R_t * \mathbb{1}(C_t \leq \hat{C}) \}. \tag{7}$$

However, directly applying eqs. (6) and (7) to estimate the optimal goals raises three issues. **First**, the standard definitions of $R_t$ and $C_t$ are the cumulative rewards and costs from the current timestep to

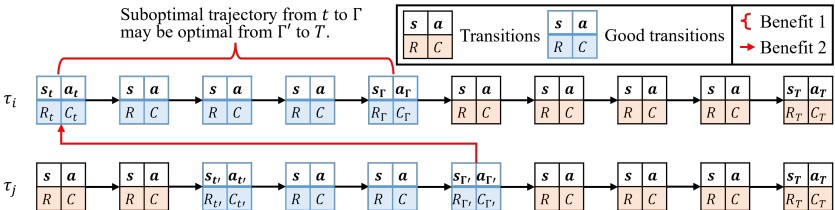

Figure 3: Necessity of temporal segmented return augmentation.

the trajectory end. However, good transitions often occur within shorter segments. This motivates our introduction of $R_{t:\Gamma}$ and $C_{t:\Gamma}$ in section 5.2, representing cumulative values over a shorter window from $t$ to $\Gamma$. **Second**, in the training stage, the target reward/cost returns are derived from cumulative values along the trajectory, which may not consistently align with the predefined targets in the testing stage. To address this problem, we propose a transition-level return relabeling in the training stage in section 5.3. **Third**, rare "lucky" transitions with high rewards and low costs can lead to value overestimation for maximum achievable goals due to transition function stochasticity. To address this problem, we design a goal function with expectile regression to estimate the expectation of the upper quantile of the return-to-go distribution in section 5.4.

## 5.2 TEMPORAL SEGMENTED RETURN AUGMENTATION

Processing the dataset in smaller temporal segments provides GM training with more abundant information to enhance cross-trajectory stitching. Specifically, we leverage such insights and augment the return information in the dataset according to:

$$(s_t, a_t, R_t, C_t) \rightarrow \{(s_t, a_t, R_{t:\Gamma}, C_{t:\Gamma}) | \Gamma = t, ..., T\} = \{(s_{t'}, a_{t'}, R_{t':T}, C_{t':T}) | t' = t + T - \Gamma\},$$

where $R_{t:\Gamma} = r_t + ... + r_\Gamma$ and $C_{t:\Gamma} = c_t + ... + c_\Gamma$. When sampling $(s_{t'}, a_{t'}, R_{t'}, C_{t'})$, GAS can seek better $R_{t:\Gamma} > R_{t'}$ & $C_{t:\Gamma} \leq C_{t'}$ for the same state from other transitions under augmentation as long as $\Gamma - t = T - t'$. As illustrated by fig. 3, this data augmentation scheme provides two benefits: (1) It substantially expands the training data by including transitions of varying temporal lengths; (2) It enables more flexible transition stitching across different timesteps by providing diverse time intervals.

## 5.3 TRANSITION-LEVEL RETURN RELABELING

First observed in Liu et al. (2023b), the misalignment between human-specified reward-cost targets in the testing stage and the training inputs can lead to degraded performance for GM methods due to their nature of behavior cloning. Inspired by the trajectory-level labeling in Liu et al. (2023b), we propose a more fine-grained, transition-level return relabeling mechanism fitting in with transition-level stitching. For sampled transitions $\hat{\mathcal{D}} = \{(s_t, a_t, R_{t:\Gamma}, C_{t:\Gamma}, t' = t + T - \Gamma)\}$, we relabel $R_{t:\Gamma}$ and $C_{t:\Gamma}$ in eq. (8) and take relabeled values as input of goal functions $V_{t:\Gamma}^R(s_t, a_t, \hat{R}_{t:\Gamma}, \hat{C}_{t:\Gamma}, t')$.

$$\begin{aligned} \hat{R}_{t:\Gamma} &= U((1 - \delta)R_{t:\Gamma}, (1 + \delta)R_{t:\Gamma}), \\ \hat{C}_{t:\Gamma} &= U(C_{t:\Gamma}, C^{\text{max}}), \end{aligned} \tag{8}$$

where $U(a, b)$ is a uniform distribution between $a$ and $b$, $\delta \in (0, 1)$ is a hyper-parameter and $C^{\text{max}}$ is the maximum value of cost returns. In this way, goal functions can be trained under more imbalanced and comprehensive reward-cost targets. Notably, our proposed GAS does not directly update the policy guided by the relabeled values. Instead, we utilize them to assist training through intermediate optimal goals and update the policy based on these goals during optimization. As a result, GAS retains the robustness of the policy without affecting reward maximization and constraint satisfaction.

## 5.4 GOAL FUNCTIONS WITH EXPECTILE REGRESSIONS

Since naively taking the maximum operator in the dataset can be prone to rare "lucky" samples, we adopt a distributional perspective and optimize goal functions that focus more on high return-to-go samples and less on low return-to-go samples. To this end, we employ expectile regression for iteratively updating the estimated goal functions.

The reward goal function should output the largest reward-to-go that satisfies the constraint. To formalize this, we first define the advantage function as:

$$A_{t:\Gamma}^R = \mathbb{1}(V_{t:\Gamma}^C < \hat{C}_{t:\Gamma}) \cdot R_{t:\Gamma} - V_{t:\Gamma}^R(s_t, \hat{R}_{t:\Gamma}, \hat{C}_{t:\Gamma}, t' = t + T - \Gamma), \tag{9}$$

where $\mathbb{1}(V_{t:\Gamma}^C < \hat{C}_{t:\Gamma})$ is an indicator function of constraint satisfaction.

In this way, transitions that violate the constraint or have low return $R_{t:\Gamma}$ are down-weighted during expectile regression. Then the loss function with expectile regression can be defined as :

$$L_R = \mathbb{E}_{\hat{D}}[|\alpha - \mathbb{1}(A_{t:\Gamma}^R < 0)| \cdot (A_{t:\Gamma}^R)^2]. \tag{10}$$

With this loss function, the reward goal function ($V_{t:\Gamma}^R$) converges to the expectile of the largest reward return-to-go that satisfies the constraint controlled by $\alpha$.

The optimization objective for the cost goal function differs from that of the reward goal function. While the reward goal function aims to find the largest reward return-to-go in the dataset via expectile regression, the cost goal function seeks to estimate the cost value associated with the optimal reward goal. To address this, we modify the loss function of the cost goal function to eqs. (11) and (12), assigning higher weights to transitions with higher reward goals under the constraint.

$$A_{t:\Gamma}^C = C_{t:\Gamma} - V_{t:\Gamma}^C(s_t, \hat{R}_{t:\Gamma}, \hat{C}_{t:\Gamma}, t' = t + T - \Gamma). \tag{11}$$

$$L_C = \mathbb{E}_{\hat{D}}[|\alpha - \mathbb{1}(A_{t:\Gamma}^R < 0)| \cdot (A_{t:\Gamma}^C)^2]. \tag{12}$$

It is worth noting that the reward and cost goal functions are not trained separately but derived under the same optimization framework with a theoretical guarantee.

**Goal-guided Policy Optimization:** To ensure that the policy comprehends optimal targets under relabeled targets returns, we incorporate the optimal reward and cost goals obtained from the goal functions as inputs to the policy. Then we optimize the policy utilizing a constrained version of Advantage Weight Regression (AWR), as shown in:

$$L_\pi = \mathbb{E}_{\hat{D}}[\mathbb{1}(V_{t:\Gamma}^C < \hat{C}_{t:\Gamma}) \cdot |\alpha - \mathbb{1}(A_{t:\Gamma}^R < 0)| \cdot (\pi(a|s_t, \hat{R}_{t:\Gamma}, \hat{C}_{t:\Gamma}, V_{t:\Gamma}^R, V_{t:\Gamma}^C, t') - a_t)^2]. \tag{13}$$

## 5.5 DATASET RESHAPING

Existing GM methods take reward and cost returns as input, while the reward-cost distribution in the dataset is often highly imbalanced, a problem referred to as data imbalance (Kang et al., 2021; Bagui & Li, 2021; Yang et al., 2021; Ren et al., 2022). The issue seriously affects the ability of RL-based methods on constraint satisfaction (Yao et al., 2024), as well as GM-assisted methods.

We define transitions with low costs and low rewards as "**conservative transitions**", transitions with high costs and high rewards as "**aggressive transitions**", and transitions with low costs but high rewards as "**ideal transitions**". As shown in fig. 4, most transitions in the dataset are concentrated in regions where both cost and reward returns are extremely low. When training with uniform sampling, GM methods tend to learn

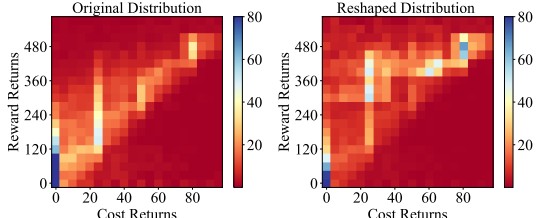

Figure 4: Original and reshaped dataset distribution.

predominantly from these conservative transitions, while under-representing ideal transitions that exhibit higher reward returns with lower cost returns. To mitigate this issue, we propose to reshape the dataset distribution during the training phase. In particular, we first estimate the reward-return distribution conditioned on cost returns from the offline dataset and then select all transitions that fall within the top $q\%$ reward returns for each cost return, thereby creating a new dataset $\mathcal{D}^q = \{(s, a, R, C) \sim \mathcal{D} | P^c(R|C) > 1 - q\}$, where $P^c$ indicates cumulative distribution function. Throughout the training procedure, $\mathcal{D}^q$ will be sampled with a probability $\epsilon$ and the original dataset $\mathcal{D}$ will be sampled with a probability $1 - \epsilon$. As shown in fig. 4, the reshaped dataset distribution is more balanced compared to the original one.

## 6 EXPERIMENT

In this section, we aim to evaluate our proposed GAS and empirically answer three questions: **1)** Can GAS achieve both safe and better performance with improved stitching ability? **2)** Can GAS

Table 1: Normalized evaluation results. The normalized cost threshold is set to 1. Values shown as "mean±std" represent the mean and standard deviation. Each value represents the average performance over 10 evaluation episodes with 5 seeds and 3 thresholds. **Bold**/gray/**blue** indicate **safe**/unsafe/**safe and best-performing** results. ↑ (↓) indicates that higher (lower) values are better.

| Methods | CPQ | | COptiDICE | | WSAC | | VOCE | | CDT | | FISOR | | CAPS | | CCAC | | GAS | |
|---|---|---|---|---|---|---|---|---|---|---|---|---|---|---|---|---|---|---|
| Tasks | R↑ | C↓ | R↑ | C↓ | R↑ | C↓ | R↑ | C↓ | R↑ | C↓ | R↑ | C↓ | R↑ | C↓ | R↑ | C↓ | R↑ | C↓ |
| Tight Constraint Threshold: Average results on thresholds with 10%, 20%, and 30% of the maximum costs for each task. | | | | | | | | | | | | | | | | | | |
| AntRun | 0.02±0.01 | 0.00±0.00 | 0.60±0.03 | 0.45±0.22 | 0.29±0.04 | 0.30±0.28 | 0.23±0.05 | 0.87±0.36 | 0.72±0.03 | 0.93±0.46 | 0.29±0.03 | 0.00±0.00 | 0.64±0.04 | 0.96±0.22 | 0.14±0.01 | 0.01±0.01 | 0.72±0.02 | 0.70±0.30 |
| BallRun | 0.32±0.14 | 1.53±1.07 | 0.58±0.01 | 4.54±0.33 | 1.10±0.30 | 7.10±0.94 | 1.08±0.38 | 7.10±0.11 | 0.33±0.01 | 1.16±0.15 | 0.23±0.01 | 0.00±0.00 | 0.15±0.04 | 1.04±0.25 | 0.77±0.00 | 3.65±1.43 | 0.33±0.04 | 0.62±0.15 |
| CarRun | 0.95±0.14 | 0.83±0.15 | 0.96±0.03 | 0.00±0.00 | 0.96±0.09 | 0.31±0.22 | 0.95±0.26 | 8.09±0.03 | 0.99±0.01 | 0.99±0.15 | 0.82±0.01 | 0.00±0.00 | 0.98±0.06 | 0.38±0.42 | 0.96±0.00 | 0.47±0.72 | 0.99±0.01 | 0.19±0.05 |
| DroneRun | 0.41±0.21 | 4.47±1.40 | 0.74±0.16 | 3.42±0.27 | 0.38±0.12 | 0.41±0.13 | 0.67±0.20 | 4.21±1.21 | 0.61±0.01 | 1.01±0.15 | 0.37±0.05 | 0.41±0.17 | 0.51±0.04 | 1.11±1.41 | 0.40±0.05 | 3.07±0.30 | 0.60±0.02 | 0.22±0.13 |
| AntCircle | 0.02±0.02 | 0.00±0.00 | 0.23±0.03 | 2.14±0.24 | 0.26±0.08 | 0.61±0.33 | 0.17±0.06 | 0.83±0.24 | 0.54±0.02 | 1.46±0.08 | 0.13±0.03 | 0.00±0.00 | 0.47±0.06 | 0.31±0.41 | 0.22±0.09 | 0.75±0.33 | 0.52±0.02 | 0.96±0.10 |
| BallCircle | 0.66±0.23 | 0.61±0.44 | 0.70±0.02 | 3.53±0.00 | 0.73±0.08 | 0.30±0.08 | 0.74±0.07 | 1.10±0.33 | 0.73±0.00 | 1.23±0.18 | 0.28±0.02 | 0.20±0.03 | 0.63±0.02 | 0.47±0.05 | 0.77±0.01 | 0.35±0.01 | 0.71±0.00 | 0.84±0.20 |
| CarCircle | 0.72±0.02 | 1.22±0.60 | 0.48±0.03 | 2.78±0.34 | 0.64±0.14 | 0.21±0.12 | 0.66±0.13 | 1.21±0.40 | 0.75±0.02 | 0.97±0.10 | 0.24±0.05 | 0.00±0.00 | 0.64±0.04 | 0.66±0.06 | 0.76±0.01 | 1.42±0.38 | 0.70±0.03 | 0.84±0.13 |
| DroneCircle | 0.05±0.02 | 2.68±1.33 | 0.41±0.02 | 1.24±0.24 | 0.02±0.01 | 0.66±0.38 | 0.05±0.02 | 1.41±0.43 | 0.69±0.01 | 1.19±0.28 | 0.49±0.03 | 0.00±0.00 | 0.64±0.02 | 0.62±0.06 | 0.35±0.04 | 0.78±0.15 | 0.68±0.01 | 0.96±0.27 |
| Average | 0.39±0.10 | 1.42±0.62 | 0.59±0.04 | 2.26±0.21 | 0.55±0.11 | 1.24±0.31 | 0.57±0.15 | 3.10±0.39 | 0.67±0.01 | 1.12±0.19 | 0.36±0.03 | 0.03±0.01 | 0.58±0.03 | 0.69±0.33 | 0.55±0.02 | 1.31±0.31 | 0.66±0.02 | 0.67±0.17 |
| PointCircle1 | 0.66±0.09 | 1.80±0.91 | 0.91±0.02 | 5.76±0.30 | 0.40±0.12 | 3.02±1.44 | 0.54±0.10 | 4.83±0.80 | 0.70±0.01 | 0.63±0.17 | 0.73±0.03 | 1.23±0.39 | 0.62±0.06 | 1.37±0.62 | 0.71±0.02 | 1.54±0.37 | 0.69±0.01 | 0.38±0.14 |
| PointCircle2 | 0.74±0.11 | 4.53±1.22 | 0.91±0.02 | 5.92±0.26 | 0.55±0.05 | 1.64±0.65 | 0.85±0.09 | 5.31±0.65 | 0.77±0.01 | 1.18±0.13 | 0.84±0.02 | 1.42±0.23 | 0.71±0.04 | 0.90±0.62 | 0.10±0.12 | 3.87±2.21 | 0.75±0.02 | 0.99±0.2 |
| CarCircle1 | 0.45±0.16 | 3.00±1.44 | 0.79±0.03 | 4.95±0.45 | 0.48±0.10 | 5.12±2.37 | 0.14±0.06 | 9.03±0.70 | 0.71±0.04 | 2.25±0.44 | 0.75±0.03 | 2.15±0.68 | 0.69±0.01 | 1.62±0.43 | 0.52±0.08 | 3.96±1.43 | 0.56±0.03 | 0.62±0.30 |
| CarCircle2 | 0.65±0.07 | 2.32±0.90 | 0.79±0.04 | 3.82±0.38 | 0.58±0.05 | 1.29±0.44 | 0.57±0.06 | 4.72±0.46 | 0.72±0.03 | 2.26±0.34 | 0.57±0.03 | 0.40±0.27 | 0.56±0.06 | 0.82±0.37 | 0.57±0.02 | 2.44±1.08 | 0.50±0.01 | 0.74±0.20 |
| Loose Constraint Threshold: Average results on thresholds with 70%, 80%, and 90% of the maximum costs for each task. | | | | | | | | | | | | | | | | | | |
| AntRun | 0.06±0.02 | 0.00±0.00 | 0.60±0.02 | 0.12±0.11 | 0.52±0.09 | 0.62±0.14 | 0.40±0.04 | 0.75±0.13 | 0.79±0.03 | 0.77±0.02 | 0.29±0.03 | 0.00±0.00 | 0.85±0.03 | 1.00±0.11 | 0.08±0.02 | 0.00±0.00 | 0.84±0.03 | 0.93±0.05 |
| BallRun | 1.20±0.00 | 1.45±0.00 | 0.57±0.01 | 0.88±0.08 | 1.21±0.33 | 1.46±0.38 | 1.14±0.01 | 1.45±0.01 | 0.72±0.02 | 0.94±0.08 | 0.23±0.01 | 0.00±0.00 | 0.47±0.10 | 1.18±0.03 | 1.21±0.00 | 1.44±0.00 | 0.76±0.00 | 0.96±0.04 |
| CarRun | 0.95±0.05 | 0.73±0.32 | 0.96±0.04 | 0.15±0.14 | 0.94±0.08 | 0.31±0.08 | 0.95±0.09 | 2.00±0.02 | 0.99±0.00 | 0.87±0.04 | 0.82±0.01 | 0.01±0.00 | 0.99±0.00 | 0.28±0.35 | 0.86±0.05 | 0.19±0.20 | 0.99±0.00 | 0.69±0.05 |
| DroneRun | 0.27±0.08 | 0.59±0.25 | 0.80±0.19 | 0.67±0.08 | 1.00±0.25 | 1.22±0.20 | 0.92±0.27 | 1.28±0.38 | 0.70±0.04 | 0.48±0.02 | 0.37±0.05 | 0.22±0.09 | 0.53±0.05 | 0.22±0.28 | 0.60±0.10 | 0.81±0.13 | 0.89±0.03 | 0.92±0.01 |
| AntCircle | 0.10±0.05 | 0.18±0.16 | 0.25±0.03 | 0.50±0.06 | 0.62±0.05 | 0.83±0.12 | 0.05±0.02 | 0.32±0.27 | 0.65±0.02 | 0.55±0.03 | 0.13±0.03 | 0.00±0.00 | 0.70±0.04 | 0.61±0.06 | 0.19±0.03 | 0.40±0.08 | 0.77±0.02 | 0.74±0.01 |
| BallCircle | 0.77±0.07 | 0.92±0.12 | 0.94±0.02 | 0.75±0.00 | 0.81±0.07 | 0.82±0.15 | 0.97±0.01 | 0.94±0.02 | 0.92±0.00 | 0.92±0.06 | 0.28±0.02 | 0.00±0.00 | 0.92±0.01 | 0.84±0.02 | 0.92±0.01 | 0.83±0.00 | 0.92±0.00 | 0.90±0.08 |
| CarCircle | 0.81±0.05 | 0.84±0.09 | 0.47±0.02 | 0.63±0.07 | 0.81±0.06 | 1.30±0.14 | 0.79±0.19 | 1.26±0.35 | 0.85±0.02 | 0.80±0.04 | 0.24±0.05 | 0.00±0.00 | 0.86±0.01 | 0.93±0.01 | 0.81±0.03 | 0.93±0.08 | 0.87±0.02 | 0.93±0.03 |
| DroneCircle | 0.02±0.00 | 0.11±0.04 | 0.41±0.02 | 0.27±0.06 | 0.03±0.01 | 0.61±0.38 | 0.05±0.01 | 0.51±0.27 | 0.79±0.02 | 0.78±0.06 | 0.49±0.03 | 0.01±0.00 | 0.85±0.03 | 0.94±0.01 | 0.21±0.03 | 0.57±0.12 | 0.87±0.03 | 0.89±0.09 |
| Average | 0.52±0.04 | 0.60±0.12 | 0.63±0.04 | 0.50±0.08 | 0.74±0.12 | 0.90±0.20 | 0.66±0.08 | 1.06±0.18 | 0.80±0.02 | 0.76±0.04 | 0.36±0.03 | 0.03±0.01 | 0.77±0.03 | 0.75±0.11 | 0.61±0.03 | 0.65±0.08 | 0.86±0.02 | 0.87±0.05 |
| PointCircle1 | 0.66±0.17 | 0.34±0.26 | 0.90±0.01 | 1.18±0.07 | 0.88±0.03 | 1.31±0.08 | 0.59±0.18 | 1.34±0.25 | 0.73±0.01 | 0.31±0.06 | 0.73±0.03 | 0.92±0.17 | 0.76±0.06 | 0.68±0.21 | 0.75±0.03 | 0.97±0.19 | 0.88±0.01 | 0.97±0.03 |
| PointCircle2 | 0.61±0.17 | 0.98±0.29 | 0.91±0.02 | 1.23±0.06 | 0.90±0.03 | 1.08±0.12 | 0.83±0.07 | 1.11±0.07 | 0.76±0.01 | 0.18±0.05 | 0.84±0.02 | 0.74±0.15 | 0.80±0.04 | 0.70±0.16 | 0.06±0.05 | 0.09±0.17 | 0.88±0.01 | 0.95±0.05 |
| CarCircle1 | 0.54±0.15 | 1.01±0.26 | 0.80±0.03 | 1.01±0.04 | 0.63±0.06 | 1.28±0.16 | 0.29±0.09 | 1.52±0.22 | 0.71±0.04 | 0.57±0.11 | 0.75±0.03 | 0.48±0.13 | 0.80±0.03 | 0.80±0.06 | 0.42±0.11 | 1.04±0.19 | 0.82±0.02 | 0.89±0.07 |
| CarCircle2 | 0.76±0.05 | 1.20±0.06 | 0.79±0.04 | 0.79±0.08 | 0.71±0.05 | 0.75±0.13 | 0.53±0.05 | 1.09±0.07 | 0.73±0.04 | 0.58±0.12 | 0.57±0.03 | 0.09±0.05 | 0.72±0.05 | 0.54±0.09 | 0.49±0.06 | 1.21±0.05 | 0.80±0.03 | 0.82±0.08 |

preserve the zero-shot adaptation ability to different constraint thresholds? **3)** Is GAS robust to imbalanced and human-specified target reward-cost return-to-goes? Accordingly, we design the following experimental setup to evaluate GAS.

**Tasks:** We evaluate GAS on the widely used **Bullet-Safety-Gym** (Gronauer, 2022) and **Safety-Gymnasium** (Ji et al., 2023) benchmarks. For all tasks, we use the dataset provided in DSRL (Liu et al., 2023a) as our offline dataset, following the D4RL (Fu et al., 2020) benchmark format.

**Baselines:** We compare our proposed GAS with the following baseline methods: (1) Constraint penalized method: CPQ (Xu et al., 2022); (2) Distribution correction estimation: COptiDICE (Lee et al., 2022); (3) Variational optimization with conservative estimation: VOCE (Guan et al., 2023); (4) Weighted safe actor-critic: WSAC (Wei et al., 2024); (5) Generative model assisted algorithms: CDT (Liu et al., 2023b) and FISOR (Zheng et al., 2024);(6) RL methods with zero-shot ability: CAPS (Chemingui et al., 2025) and CCAC (Guo et al., 2025).

**Metrics:** We evaluate performance using normalized reward and cost returns. The normalized reward return is defined by $R_{\text{norm}} = \frac{R_\pi}{R_{\max}}$, where $R_\pi$ is the reward return achieved by policy $\pi$ and and $R_{\max}$ is the maximum reward return in the dataset. The normalized cost return is defined by $C_{\text{norm}} = \frac{C_\pi}{L}$, where $C_\pi$ is the cost return achieved by policy $\pi$ and $L$ is the selected threshold. To provide better interpretation to decouple the two objectives in OSRL compared to traditional fixed thresholds, we use percentage-based thresholds calibrated to each task's cost range. Specifically, we define tight constraints as 10%, 20%, and 30% of the maximum cost to emphasize constraint satisfaction, and loose constraints as 70%, 80%, and 90% of the maximum cost to focus on reward maximization.

## 6.1 CAN GAS ACHIEVE BOTH SAFE AND BETTER PERFORMANCE WITH IMPROVED STITCHING ABILITY?

The experiment results of different baselines under various tasks on tight and loose constraints are presented in table 1. In tight constraint settings, only GAS achieves the best and safe performance in all tasks. Traditional RL methods, such as CPQ, COptiDICE, WSAC, and VOCE, suffer from severe constraint violations both on average and for each task. Even among GM methods, CDT, while outperforming RL-assisted methods, still fails to ensure constraint satisfaction in multiple scenarios where GAS succeeds. FISOR maintains safety in most tasks but at a substantial cost to performance, with rewards significantly lower than GAS. In such tight constraint settings, the superiority of GAS over CDT on constraint satisfaction comes from the stitching ability, where GAS can stitch safe transitions among different timesteps and trajectories together. In loose constraint settings, although most baselines exhibit safe performance, GAS achieves the best performance on reward maximization. This performance gap between GAS and CDT highlights GAS's advanced reward maximization

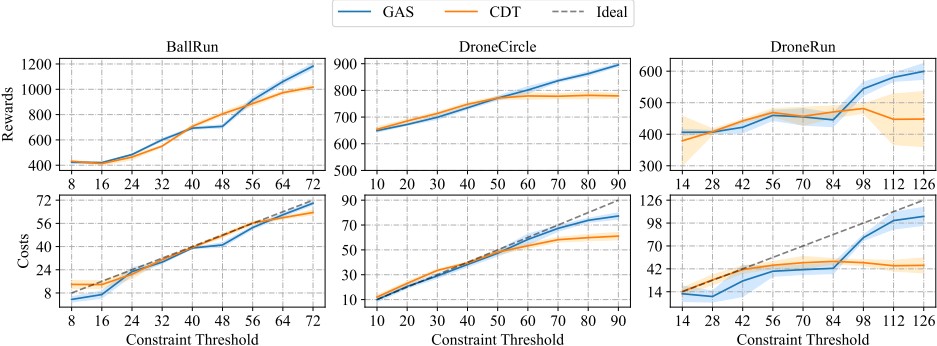

Figure 5: Evaluation results on zero-short adaptation. The x-axis indicates different selected thresholds and the y-axis indicates corresponding performance on cumulative rewards and costs. "Ideal" line indicates the case when the cumulative costs are equal to the constraint thresholds.

capabilities, leveraging its innovative transition stitching approach to combine high-reward segments while maintaining robust safety guarantees.

## 6.2 CAN GAS PRESERVE THE ZERO-SHOT ADAPTATION ABILITY TO DIFFERENT CONSTRAINT THRESHOLDS?

A critical advantage of GM methods is their zero-shot adaptation ability in handling different constraint thresholds without retraining. To validate this ability in GAS, we compare our method with CDT and test thresholds from $10\%$ to $90\%$ of the maximum costs, increased by $10\%$ each time. The results are shown in fig. 5. Compared with CDT, the cumulative costs of GAS are consistently below the constraint thresholds, whereas CDT performs unsafely in some cases, especially when the constraint thresholds are smaller than $30\%$ of the maximum costs. In addition, as the constraint threshold increases, GAS flexibly adjusts its behavior and achieves progressively higher rewards while ensuring safety, but CDT tends to be overly conservative.

## 6.3 IS GAS ROBUST TO IMBALANCED AND HUMAN-SPECIFIED TARGET REWARD-COST RETURN-TO-GOES?

To demonstrate GAS's robustness against imbalanced target reward-cost return-to-goes, we compare GAS under different return relabeling levels (denoted as "GAS-$\delta$" and "GAS w/o Relabel") in settings with the constraint threshold $20\%$ of the maximum costs and varying target reward return-to-goes. As shown in fig. 6, GAS consistently achieves safe performance as the target reward return-to-go increases. In contrast, GAS w/o Relabel can only achieve safe performance for a narrow band of reward targets. As $\delta$ increases, GAS becomes more robust against imbalanced targets. This highlights one of the GAS's key innovations: the transition-level return relabeling method in enhancing robustness.

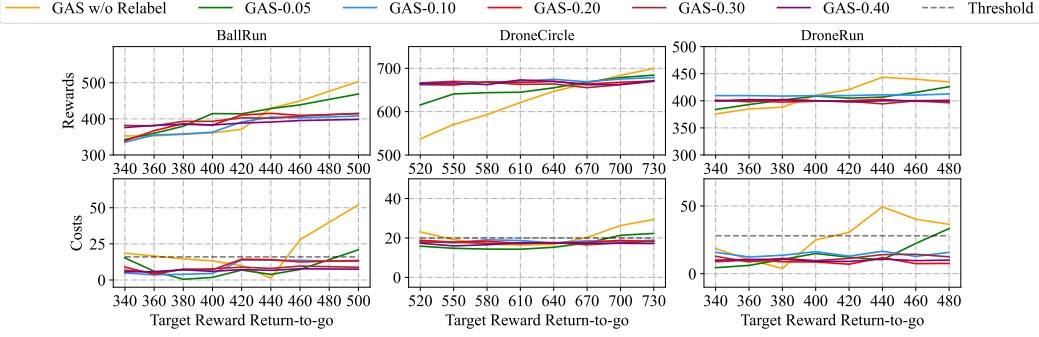

Figure 6: Evaluation results on robustness of imbalanced target reward-cost return-to-goes. The x-axis indicates different target reward return-to-go and the y-axis indicates corresponding performance on cumulative rewards and costs. "Threshold" line indicates constraint thresholds.

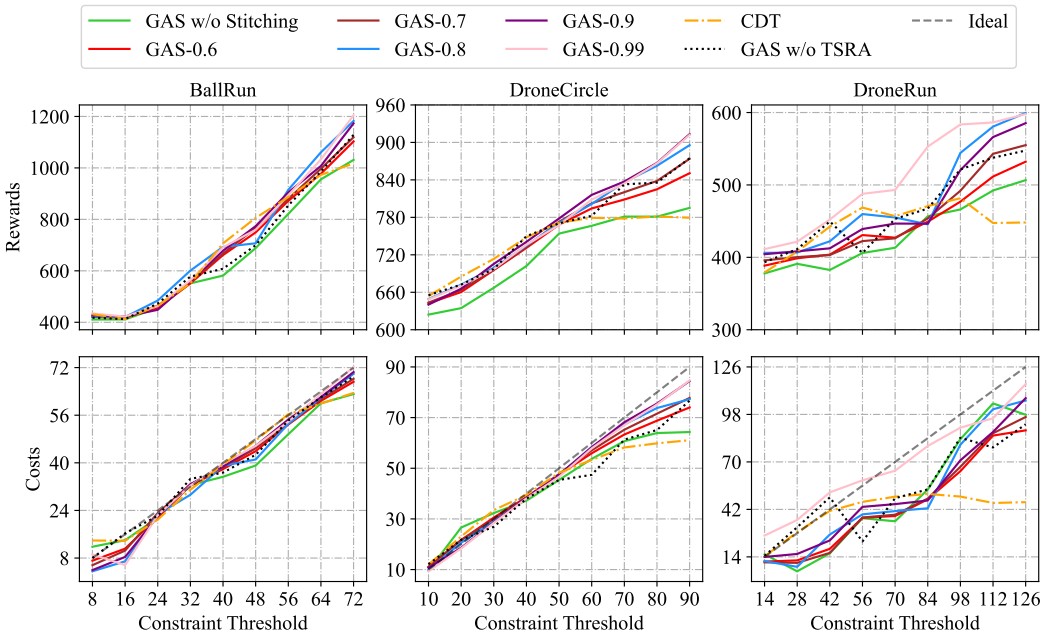

Figure 7: Ablation studies on the stitching ability (expectile regression) and temporal segmented return augmentation.

## 6.4 ABLATION STUDY ON STITCHING ABILITY AND TSRA

To verify the theoretical improvement for GAS's component in section B, we compare GAS under different expectile level $\alpha$ from $0.5$ to $0.99$ (where $\alpha = 0.5$ is denoted as "GAS w/o Stitching" and others are denoted as GAS-$\alpha$), and a variant without Temporal Segmented Return Augmentation (denoted as "GAS w/o TSRA") in fig. 7. As shown in fig. 7, without expectile regression, GAS w/o Stitching degrades to conditional behavior cloning ($\alpha = 0.5$), which has similar performance and problems to CDT. This result is also consistent with section 4.1, where both GAS w/o Stitching and CDT suffer from insufficient stitching ability. If $\alpha$ is extremely high, GAS-0.99 fails to keep safety in DroneRun tasks. This result is consistent with our claims of the problem of "lucky" biases in section 5.1. As $\alpha$ increase from $0.6$ to $0.9$, GAS is well-performed and demonstrates robustness on $\alpha$. The performance of GAS w/o TSRA is almost completely inferior to GAS. This result validates the theoretical analysis result, where the optimality without augmentation is just a lower bound of that with augmentation. Furthermore, GAS w/o TSRA suffers from constraint violation for tight constraint thresholds compared to GAS. This result indicates that the flexible sub-trajectories augmented by TSRA provide additional benefits when original trajectories fail to satisfy the tight constraint thresholds.

## 7 CONCLUSION

Aiming to address the limitations of existing GM-assisted OSRL methods, we propose a novel algorithm named GAS that concentrates on enhancing the stitching ability for a better balance of reward maximization and constraint satisfaction. GAS focuses on achieving better reward maximization and constraint satisfaction by training specialized reward and cost goal functions via expectile regression. These goal functions estimate the optimal achievable reward and cost returns and are used to guide the policy in effectively stitching transitions under relabeled reward-cost return-to-goes. Experiment results demonstrate GAS's superiority on reward maximization, constraint satisfaction, zero-shot adaptation, and robustness on imbalanced target reward-cost return-to-goes. A potential weakness of GAS is the trade-off between memory capability and stitching ability, as the current attention module struggles to fully capture true temporal dependencies in CMDPs. Besides, due to the limitation of the existing goal-conditioned framework, GAS's stitching ability focuses on improving performance by better selecting high-quality sub-segments in the safe dataset, rather than constructing new safe trajectories by combining safe and unsafe data in RL. Future work could address this limitation by designing more advanced memory mechanisms and frameworks into GAS to further enhance the robustness and performance of GM-assisted policies.

## ACKNOWLEDGEMENT

This work was supported by the Hong Kong Research Grants Council under the Areas of Excellence scheme grant AoE/E-601/22-R and NSFC/RGC Collaborative Research Scheme grant CRS_HKUST603/22.

## REPRODUCIBILITY STATEMENT

This paper introduces a new algorithm, named GAS. A clear theoretical explanation of GAS is illustrated in sections A and B. The clear pseudo code is shown in algorithm 1. Detailed experiment settings, including hyperparameters, are shown in section E. The experimental code is available at `https://github.com/Ziffer-byakuya/GAS`.

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

## A  EXPECTILE REGRESSION

Expectile regression (Koenker & Hallock, 2001) is a statistical modeling technique that generalizes traditional mean regression to obtain the weighted means across different parts of the distribution. Given a set of samples $\{x_i | i = 1, ..., N\}$ under a distribution, the expectile regression aims to achieve eq. (14) where $\alpha$ is the expectile level and $u = x - \bar{x}^\alpha$.

$$\min_{\bar{x}^\alpha} \mathbb{E}[|\alpha - \mathbb{1}(u < 0)| \cdot u^2] \to \min_{\bar{x}^\alpha} \mathbb{E}[|\alpha - \mathbb{1}((x - \bar{x}^\alpha) < 0)| \cdot (x - \bar{x}^\alpha)^2], \tag{14}$$

**Property 1 (Kostrikov et al., 2022):**

As $\alpha$ increases from 0.5 to 1, $\bar{x}^\alpha$ increases from the mean value to the largest value of a distribution:

$$\lim_{\alpha \to 1} \bar{x}^\alpha = \max_{x_i}\{x_i | i = 1, ..., N\} \tag{15}$$

For example, if $\alpha = 0.5$, it is the same as MSE since $|0.5 - \mathbb{1}(.)| = 0.5$; if $\alpha = 0.9$, it gives a weight of 0.9 to samples $x \geq \bar{x}^\alpha$ but only gives a weight of 0.1 to samples $x < \bar{x}^\alpha$.

This property makes expectile regression widely used in the estimation of the optimal value function in reinforcement learning (Kostrikov et al., 2021; Dabney et al., 2018) since the optimal value function is naturally defined as the largest reward return in a task.

**Property 2 (Hansen-Estruch et al., 2023):**

Let us represent the expectile regression as $f(u) = |\alpha - \mathbb{1}(u < 0)| \cdot u^2$ and $f'(u) = \frac{df(u)}{du}$ for easy and clear writing. We have:

$$f'(u) = |f'(u)| \cdot \frac{u}{|u|} \tag{16}$$

where $f(u)$ is convex and $f'(0) = 0$.

## B  THEORETICAL ANALYSIS OF GAS

### B.1  ALGORITHM DERIVATION

Fowllowing the eq. (15) in **property 1**, eq. (9), and eq. (10), we can directly have:

$$\lim_{\alpha \to 1} V_{t:\Gamma}^R(s_t, \hat{R}_{t:\Gamma}, \hat{C}_{t:\Gamma}, t' = t + T - \Gamma) = \max_{\{(s_t, a_t, R_{t:\Gamma}, C_{t:\Gamma}, t' = t + T - \Gamma)\}} (R_{t:\Gamma} | V_{t:\Gamma}^C < \hat{C}_{t:\Gamma}) \tag{17}$$

This indicates that the reward goal function finally converges to the largest reward return-to-go within the constraint.

Then consider $L_R$ in eq. (10) as $f(A_{t:\Gamma}^R)$. When the reward goal function converges, we have

$$\begin{aligned}
\frac{\partial L_R}{\partial V_{t:\Gamma}^R} &= -\int_a \mu(a|s) \cdot |f'(A_{t:\Gamma}^R)| \cdot \frac{A_{t:\Gamma}^R}{|A_{t:\Gamma}^R|} \\
&= -\int_a \mu(a|s) \cdot \frac{|f'(A_{t:\Gamma}^R)|}{|A_{t:\Gamma}^R|} \cdot A_{t:\Gamma}^R \\
&= -\int_a \pi(a|s) \cdot A_{t:\Gamma}^R \\
&= 0
\end{aligned} \tag{18}$$

where $\pi(a|s) = \mu(a|s) \cdot \frac{|f'(A_{t:\Gamma}^R)|}{N^{scale}|A_{t:\Gamma}^R|}$ is the target policy.

With the target policy, we can define the loss function of the cost goal function as $L_C = E_\pi[(C_{t:\Gamma} - V_{t:\Gamma}^C)^2]$. When it converges, we have:

$$
\begin{aligned}
0 &= -2E_\pi[C_{t:\Gamma} - V_{t:\Gamma}^C] \\
&= -2\int_a \pi(a|s) \cdot (C_{t:\Gamma} - V_{t:\Gamma}^C) \\
&= -2/N^{scale} \cdot \int_a \mu(a|s) \cdot |\alpha - \mathbb{1}(A_{t:\Gamma}^R < 0)| \cdot (C_{t:\Gamma} - V_{t:\Gamma}^C)
\end{aligned}
\tag{19}
$$

Thus, we can obtain the loss function of the cost goal function as eq. (12).

During the policy extraction, the target policy is supposed to be $\pi(a|s) = \mu(a|s) \cdot \frac{|f'(A_{t:\Gamma}^R)|}{N^{scale}|A_{t:\Gamma}^R|}$. However, to further ensure safety, we add the constraint to the policy extraction:

$$
\pi(a|s) = 1/N^{scale} \cdot \mathbb{1}(V_{t:\Gamma}^C \le \hat{C}_{t:\Gamma}) \cdot |\alpha - \mathbb{1}(A_{t:\Gamma}^R < 0)| \cdot \mu(a|s)
\tag{20}
$$

Thus, the loss function of the target policy is eq. (13).

## B.2 ABLATION ANALYSIS

This ablation analysis aims to study the contribution of each separate component of GAS theoretically.

### B.2.1 STITCHING ABILITY (EXPECTILE REGRESSION)

Based on **property 1** in section A, it is easy to prove that:

$$
\lim_{\alpha=0.5} V_{t:\Gamma}^R(s_t, \hat{R}_{t:\Gamma}, \hat{C}_{t:\Gamma}, t') < \lim_{\alpha \to 1} V_{t:\Gamma}^R(s_t, \hat{R}_{t:\Gamma}, \hat{C}_{t:\Gamma}, t') = \max_{\{(s_t, a_t, R_{t:\Gamma}, C_{t:\Gamma}, t')\}} (R_{t:\Gamma}|V_{t:\Gamma}^C < \hat{C}_{t:\Gamma})
\tag{21}
$$

Thus the optimality guarantee is lost without expectile regression.

### B.2.2 TEMPORAL SEGMENTED RETURN AUGMENTATION

We can partition the dataset after augmentation as $D = D^o \cup D^a$, where $D^o$ is the original dataset and $D^a$ is the augmented part.

$$
\begin{aligned}
\lim_{\alpha \to 1} E_D[V_{t:\Gamma}^R(s_t, \hat{R}_{t:\Gamma}, \hat{C}_{t:\Gamma}, t')] &= \max_{\{(s_t, a_t, R_{t:\Gamma}, C_{t:\Gamma}, t') \sim D\}} (R_{t:\Gamma}|V_{t:\Gamma}^C < \hat{C}_{t:\Gamma}) \\
&= \max[\max_{\{D^o\}}(R_{t:\Gamma}|V_{t:\Gamma}^C < \hat{C}_{t:\Gamma}), \ \max_{\{D^a\}}(R_{t:\Gamma}|V_{t:\Gamma}^C < \hat{C}_{t:\Gamma})] \\
&\ge \max_{\{(s_t, a_t, R_{t:\Gamma}, C_{t:\Gamma}, t') \sim D^o\}} (R_{t:\Gamma}|V_{t:\Gamma}^C < \hat{C}_{t:\Gamma})
\end{aligned}
\tag{22}
$$

This result indicates that the optimality guarantee in the original dataset is just a lower bound of that after augmentation. With augmentation, the optimality guarantee is always larger than that in the original dataset, which further improves the stitching ability.

### B.2.3 TRANSITION-LEVEL RETURN RELABELING

Since the purpose of this method is to make GAS more robust, this paper will discuss it from the perspective of the input and output. For simplicity, the influence on the reward goal function is analyzed as an example, considering that it has no difference from that on the cost goal function and the policy.

Without Relabeling, both inputs and loss functions utilize $(R, C)$ pairs following the dataset distribution. This indicates that only with the appropriate $R$ and $C$, the goal functions and policy can achieve the optimal value:

$$
\lim_{\alpha \to 1} V_{t:\Gamma}^R(s_t, R_{t:\Gamma}, C_{t:\Gamma}, t') = \max_{\{(s_t, a_t, R_{t:\Gamma}, C_{t:\Gamma}, t')\}} (R_{t:\Gamma}|V_{t:\Gamma}^C < C_{t:\Gamma})
\tag{23}
$$

However, during the test stage, the targets $\hat{R}$ and $\hat{C}$ cannot be accurately known and need to be specified by users. Thus it will suffers from inaccurate $R$ and $C$ signals and cannot determine which target should be prioritized when $R$ and $C$ conflict with each other.

With Relabeling, the targets are relabeled to consider a more robust input distribution while loss functions still utilize true $(R, C)$ pairs to update the outputs with expectile regression. This indicates that even with inaccurate $R$ and $C$, the goal functions can still obtain the optimal value, and the policy can provide the corresponding action:

$$\lim_{\alpha \to 1} V_{t:\Gamma}^R(s_t, \hat{R}_{t:\Gamma}, \hat{C}_{t:\Gamma}, t') = \max_{\{(s_t, a_t, R_{t:\Gamma}, C_{t:\Gamma}, t')\}} (R_{t:\Gamma} | V_{t:\Gamma}^C < \hat{C}_{t:\Gamma}) \tag{24}$$

## C   ALGORITHM DETAILS

We present the full algorithm of our method in algorithm 1.

---

**Algorithm 1** Goal-Assisted Stitching (GAS)

---

1: **Network:** Initialize two goal functions $V_{t:\Gamma}^R(s, R, C, t' = t+T-\Gamma), V_{t:\Gamma}^C(s, R, C, t' = t+T-\Gamma)$, and policy $\pi(a|s, R, C, V^R, V^C, t' = t+T-\Gamma)$.
2: **for** iteration$= 0, ..., N$ **do**
3:    Sample transitions $\hat{\mathcal{D}} = \{(s, a, R_{t:\Gamma}, C_{t:\Gamma})\} \sim \mathcal{D}$ with $1-\epsilon$ and $\mathcal{D}^q$ with $\epsilon$ probability.
4:    Get augmented return and cost return:
5:        $\hat{R}_{t:\Gamma} = U((1-\delta)R_{t:\Gamma}, (1+\delta)R_{t:\Gamma})$.
6:        $\hat{C}_{t:\Gamma} = U(C_{t:\Gamma}, C^{\max})$.
7:    Get advantage function:
8:        $A_{t:\Gamma}^R = \mathbb{1}(V_{t:\Gamma}^C < \hat{C}_{t:\Gamma}) \cdot R_{t:\Gamma} - V_{t:\Gamma}^R(s_t, \hat{R}_{t:\Gamma}, \hat{C}_{t:\Gamma}, t' = t+T-\Gamma)$.
9:        $A_{t:\Gamma}^C = C_{t:\Gamma} - V_{t:\Gamma}^C(s_t, \hat{R}_{t:\Gamma}, \hat{C}_{t:\Gamma}, t' = t+T-\Gamma)$.
10:   **Update reward goal function:**
11:       $L_R = \mathbb{E}_{\hat{\mathcal{D}}}[|\alpha - \mathbb{1}(A_{t:\Gamma}^R < 0)| \cdot (A_{t:\Gamma}^R)^2]$.
12:   **Update cost goal function:**
13:       $L_C = \mathbb{E}_{\hat{\mathcal{D}}}[|\alpha - \mathbb{1}(A_{t:\Gamma}^R < 0)| \cdot (A_{t:\Gamma}^C)^2]$.
14:   **Policy Extraction:**
15:       $L_\pi = \mathbb{E}_{\hat{\mathcal{D}}}[\mathbb{1}(V_{t:\Gamma}^C < \hat{C}_{t:\Gamma}) \cdot |\alpha - \mathbb{1}(A_{t:\Gamma}^R < 0)| \cdot (\pi(a|s_t, \hat{R}_{t:\Gamma}, \hat{C}_{t:\Gamma}, V_{t:\Gamma}^R, V_{t:\Gamma}^C, t') - a_t)^2]$.

---

## D   RETURN RELABELING TECHNIQUE DETAILS

The misalignment problem between the training and test phases is first proposed in Liu et al. (2023b), which indicates that users may select target reward-cost pairs different from the training inputs. This misalignment problem has become serious when users select imbalanced target reward-cost pairs, such as extremely large target rewards and extremely small target costs. To address this issue, CDT (Liu et al., 2023b) proposes a trajectory-level return relabel method as CDT only learns policy within the trajectory. This method, although it improves the robustness of CDT, degrade the ability to maximize rewards and satisfy constraints, as the relabeled returns provide wrong information about the trajectory. Inspired by the trajectory-level return labeling in Liu et al. (2023b), we propose a more fine-grained, transition-level return relabeling mechanism fitting in with transition-level stitching. Different from CDT, our proposed GAS does not directly update the policy guided by only the relabeled values. Instead, we utilize them to assist training through intermediate optimal goals and update the policy based on these goals in the loss function during optimization. As a result, GAS retains the robustness of the policy without affecting reward maximization and constraint satisfaction.

## E   EXPERIMENT DETAILS

### E.1   BENCHMARK AND TASKS

Bullet-safety-gym (Gronauer, 2022) and Safety-gamnasium (Ji et al., 2023) are utilized for experiments. We consider 8 cases in Bullet-safety-gym and 4 cases in Safety-gamnasium, involving two tasks (*Run* and *Circle*) and multiple types of robots (*Ant*, *Ball*, *Car*, *Drone*, and *Point*). The tasks *AntCircle*, *PointCircle1*, *PointCircle2*, *CarCircle1*, and *CarCircle2* are considered as complex tasks with episode length $T = 500$ and maximum cost $C^{\max} \geq 200$, while other tasks are regarded as

Table 2: Benchmark details. The cost range indicates the maximum cumulative costs among the offline trajectories. Offline trajectories indicate the number of trajectories in the offline dataset.

| Benchmarks | Task | State Space | Action Space | Cost Range | Episode Length (T) | Offline Trajectories |
|---|---|---|---|---|---|---|
| | AntRun | 33 | 8 | 150 | 200 | 1816 |
| | BallRun | 7 | 2 | 80 | 100 | 940 |
| | CarRun | 7 | 2 | 40 | 200 | 651 |
| Bullet-safety-gym | DroneRun | 17 | 4 | 140 | 200 | 1990 |
| | AntCircle | 34 | 8 | 200 | 500 | 5728 |
| | BallCircle | 8 | 2 | 80 | 200 | 886 |
| | CarCircle | 8 | 2 | 100 | 300 | 1450 |
| | DroneCircle | 18 | 4 | 100 | 300 | 1923 |
| | PointCircle1 | 28 | 2 | 200 | 500 | 1098 |
| Safety-gymnasium | PointCircle2 | 28 | 2 | 300 | 500 | 895 |
| | CarCircle1 | 40 | 2 | 250 | 500 | 1271 |
| | CarCircle2 | 40 | 2 | 400 | 500 | 940 |

simpler tasks with episode length less than 301. Details of the parameter settings for each tasks are shown in table 2, which is part of DSRL (Liu et al., 2023a). As for the cost definition and reward definition, we follow the same standard with DSRL.

## E.2 HYPERPARAMETER

Table 3: Hyperparameters of GAS.

| Parameter | Value | Parameter | Value |
|---|---|---|---|
| Number of layers | 7 | Hidden size | 128 |
| Embedding size | 64 | Batch Size | 2048 |
| Learning rate | 0.0001 | Adam betas | (0.9,0.999) |
| Grad norm clip | 0.25 | Weight decay | 0.0001 |
| Expectile level $\alpha$ | 0.8 | Reward relabel level $\delta$ | 0.1 |
| Dataset reshape threshold $q\%$ | 10% | Sample probability $\epsilon$ | 0.5 |

The table 3 shows the detailed hyperparameter for GAS used in section 6 and table 4 shows the target reward and cost return-to-go pairs used in the testing stage. Notably, GAS is not sensitive to the target reward-cost return-to-go pairs as we demonstrate in section 6.3.

Table 4: Target reward and cost return-to-go pairs for GAS in testing stage.

| Benchmark | Task | Cost range | 10% | 20% | 30% | 70% | 80% | 90% |
|---|---|---|---|---|---|---|---|---|
| | AntRun | 150 | (690, 15) | (690, 30) | (700, 45) | (750, 105) | (800, 120) | (820, 135) |
| | BallRun | 80 | (420, 8) | (420, 16) | (500, 24) | (900, 56) | (1000, 64) | (1200, 72) |
| | CarRun | 40 | (572, 4) | (572, 8) | (572, 12) | (572, 28) | (572, 32) | (572, 36) |
| Bullet-safety-gym | DroneRun | 140 | (400, 14) | (420, 28) | (45, 42) | (600, 98) | (620, 112) | (640, 126) |
| | AntCircle | 200 | (160, 20) | (200, 40) | (240, 60) | (320, 140) | (350, 160) | (400, 180) |
| | BallCircle | 80 | (500, 8) | (600, 16) | (690, 24) | (800, 56) | (810, 64) | (820, 72) |
| | CarCircle | 100 | (370, 10) | (390, 20) | (410, 30) | (480, 70) | (480, 80) | (480, 90) |
| | DroneCircle | 100 | (600, 10) | (650, 20) | (700, 25) | (830, 70) | (850, 80) | (870, 90) |
| | PointCircle1 | 200 | (43,20) | (45,40) | (46,60) | (52,140) | (54,160) | (58,180) |
| Safety-gymnasium | PointCircle2 | 300 | (36,30) | (41,60) | (42,90) | (45,210) | (47,240) | (48,270) |
| | CarCircle1 | 250 | (4,25) | (8,50) | (10,75) | (13,175) | (15,200) | (18,225) |
| | CarCircle2 | 400 | (8,40) | (14,80) | (15,120) | (22,280) | (23,320) | (27,360) |

## E.3 MORE EXPERIMENTS ON IMPROVED STITCHING ABILITY

**Detailed results on each threshold.** We provide more detailed results on each threshold for all baselines under Bullet-safety-gym for reference in fig. 8.

**Additional results on velocity tasks.** As shown in table 5, we compare GAS with other baselines on 5 additional velocity tasks in Safety-gamnasium.

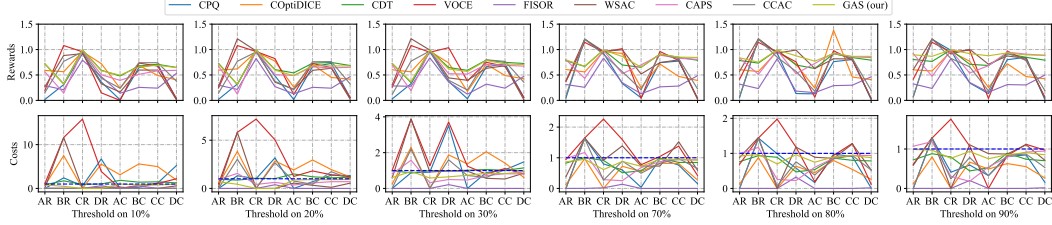

Figure 8: Normalized evaluation results on each threshold for all baselines under Bullet-safety-gym.

Table 5: Additional normalized evaluation results on Velocity tasks.

| Methods | CDT | | CAPS | | CCAC | | GAS | |
|---|---|---|---|---|---|---|---|---|
| Tasks | R ↑ | C ↓ | R ↑ | C ↓ | R ↑ | C ↓ | R ↑ | C ↓ |
| Tight Constraint Threshold. | | | | | | | | |
| HopperVelocity | $0.75_{\pm0.09}$ | $0.89_{\pm0.21}$ | $0.47_{\pm0.14}$ | $0.67_{\pm0.26}$ | $0.06_{\pm0.04}$ | $0.36_{\pm0.27}$ | $0.80_{\pm0.02}$ | $0.90_{\pm0.18}$ |
| HalfCheetahVelocity | $0.99_{\pm0.01}$ | $0.25_{\pm0.11}$ | $0.94_{\pm0.02}$ | $0.78_{\pm0.16}$ | $0.80_{\pm0.04}$ | $0.96_{\pm0.03}$ | $0.97_{\pm0.01}$ | $0.28_{\pm0.04}$ |
| SwimmerVelocity | $0.67_{\pm0.02}$ | $1.06_{\pm0.11}$ | $0.46_{\pm0.10}$ | $2.26_{\pm1.99}$ | $0.20_{\pm0.02}$ | $1.20_{\pm0.24}$ | $0.68_{\pm0.00}$ | $0.93_{\pm0.07}$ |
| Walker2dVelocity | $0.78_{\pm0.02}$ | $0.12_{\pm0.19}$ | $0.75_{\pm0.08}$ | $0.76_{\pm0.30}$ | $0.23_{\pm0.19}$ | $0.92_{\pm0.70}$ | $0.77_{\pm0.07}$ | $0.75_{\pm0.26}$ |
| AntVelocity | $0.99_{\pm0.01}$ | $0.52_{\pm0.13}$ | $0.97_{\pm0.01}$ | $0.62_{\pm0.14}$ | $0.00_{\pm0.00}$ | $0.00_{\pm0.00}$ | $0.98_{\pm0.00}$ | $0.98_{\pm0.06}$ |
| Loose Constraint Threshold. | | | | | | | | |
| HopperVelocity | $0.70_{\pm0.07}$ | $0.27_{\pm0.07}$ | $0.24_{\pm0.14}$ | $0.42_{\pm0.18}$ | $0.02_{\pm0.02}$ | $0.04_{\pm0.04}$ | $0.93_{\pm0.04}$ | $0.94_{\pm0.04}$ |
| HalfCheetahVelocity | $0.99_{\pm0.01}$ | $0.13_{\pm0.04}$ | $0.98_{\pm0.02}$ | $0.67_{\pm0.22}$ | $0.81_{\pm0.08}$ | $0.95_{\pm0.06}$ | $1.00_{\pm0.01}$ | $0.71_{\pm0.05}$ |
| SwimmerVelocity | $0.57_{\pm0.04}$ | $0.37_{\pm0.09}$ | $0.35_{\pm0.15}$ | $0.60_{\pm0.45}$ | $0.19_{\pm0.04}$ | $0.24_{\pm0.06}$ | $0.64_{\pm0.03}$ | $0.67_{\pm0.11}$ |
| Walker2dVelocity | $0.78_{\pm0.01}$ | $0.01_{\pm0.02}$ | $0.80_{\pm0.14}$ | $0.74_{\pm0.13}$ | $0.00_{\pm0.00}$ | $0.00_{\pm0.00}$ | $0.89_{\pm0.03}$ | $0.88_{\pm0.04}$ |
| AntVelocity | $0.98_{\pm0.01}$ | $0.13_{\pm0.05}$ | $0.99_{\pm0.02}$ | $0.43_{\pm0.20}$ | $0.00_{\pm0.00}$ | $0.00_{\pm0.00}$ | $0.98_{\pm0.01}$ | $0.97_{\pm0.03}$ |

## E.4 MORE EXPERIMENTS ON THE ZERO-SHOT ADAPTATION ABILITY

The fig. 9 demonstrates the zero-shot adaptation ability comparison of GAS and CDT on more complex tasks. Even on more complex tasks, GAS can preserve the reward maximization and constraint satisfaction ability for all cost ranges. CDT, in contrast, suffers from constraint violation in tight constraint settings or is overly conservative as the threshold increases.

## E.5 ABLATION STUDY ON DATASET RESHAPING METHOD

To assess the importance of the dataset reshaping on the training stability and efficiency, we compare GAS under two hyperparameters, including dataset reshape threshold $q\%$ (GAS-$q\%$) in fig. 10, and sample probability $\epsilon$ (GAS-$\epsilon$) in fig. 11, on *DroneCircle* task under different constraint thresholds. In these two figures, GAS w/o DR represents GAS without dataset reshaping, which is the same as the case when $q\% = 100\%$ or $\epsilon = 0$. In fig. 10, GAS is robust to the dataset reshape threshold when $q\% \leq 40\%$ and is sensitive when $q\%$ is extremely large. In fig. 11, the sample probability $\epsilon$ mainly influences the learning efficiency in the early training stage without affecting the final results. These results demonstrate that GAS is robust under different $q\%$ and $\epsilon$ of dataset reshaping.

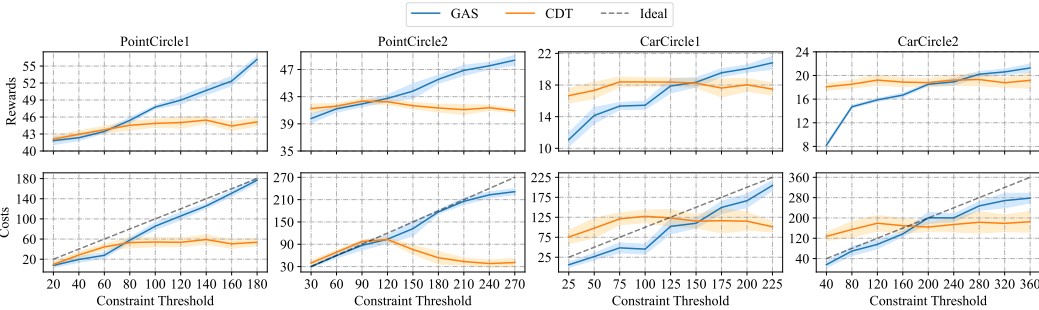

Figure 9: Evaluation results on zero-shot adaptation with complex tasks. The x-axis indicates different selected thresholds and the y-axis indicates corresponding performance on cumulative rewards and costs. "Ideal" line indicates the case when the cumulative costs are equal to the constraint thresholds.

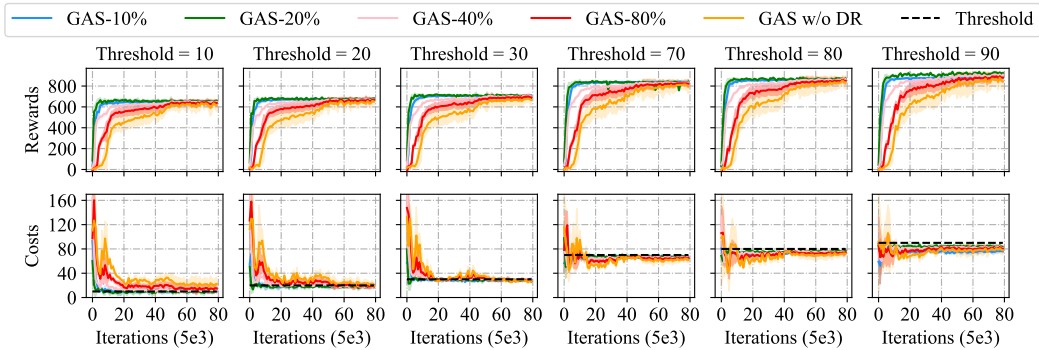

Figure 10: Hyperparameter analysis on dataset reshape threshold $q\%$ on task *DroneCircle*.

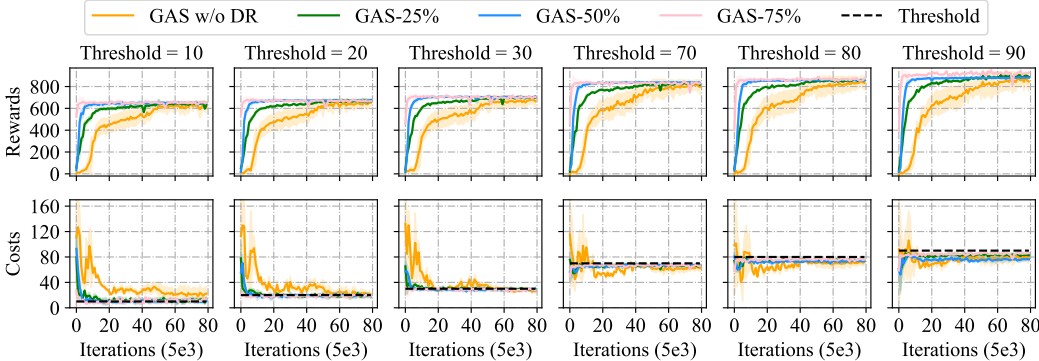

Figure 11: Hyper-parameter analysis on sample probability $\epsilon$ on task *DroneCircle*.

### E.6 INFRASTRUCTURE AND TIME COST

We run all the experiments and all the baselines with:

1. GPT version: NVIDIA GeForce RTX 3080.
2. CPU version: Intel(R) Xeon(R) Platinum 8375C CPU @ 2.90GHz.

Table 6: Training iterations and time for GAS and baselines on task *CarCircle*.

| Method | CPQ | COptiDICE | WSAC | VOCE | CDT | FISOR | GAS |
|---|---|---|---|---|---|---|---|
| Iteration times | 4e5 | 4e5 | 3e4 | 4e4 | 4e5 | 4e5 | 4e5 |
| Time | 2h19min | 2h43min | 3h10min | 3h38min | 11h32min | 1h44min | 8h20min |

We also test the training time of GAS and baselines on task *CarCircle*, as shown in table 6. For most baselines, we follow the same iteration times. As for WSAC and VOCE, we follow the training standard of the original paper and make sure their convergence since training $4e5$ times is much too long for them. Notably, both GAS and CDT only need to be trained once for all thresholds, even though the training time is much longer, while other baselines need to be re-trained for each threshold. Besides, GAS is faster than CDT as GAS only utilizes the MLP architecture, but CDT utilize the Transformer architecture.

## F THE USE OF LARGE LANGUAGE MODELS (LLMS)

In this paper, LLMs are utilized to polish writing (e.g., grammar, spelling, word choice).

