# OpenReview forum: "GAS: Enhancing Reward-Cost Balance of Generative Model-assisted Offline Safe RL"
_ICLR.cc/2026/Conference — ICLR 2026 Poster_

### Official Review · Reviewer_aKUM · 2025-10-16

**Soundness:** 3
**Presentation:** 3
**Contribution:** 2
**Rating:** 4
**Confidence:** 5

**Summary:**

This paper proposes GAS, a method designed to balance the reward and cost conditions in offline safe reinforcement learning. GAS achieves a favorable trade-off between safety and reward by combining techniques such as return augmentation, return relabeling, goal generation, and dataset reshaping.

**Strengths:**

- The paper adopts a reasonable approach to tackling offline safe reinforcement learning by focusing on balancing reward and cost.
- The use of goal generation to address the reward–cost trade-off is well-motivated.
- The proposed method demonstrates strong reward performance in tasks with relatively low safety requirements.

**Weaknesses:**

- Methodology
    - The proposed method combines return augmentation, return relabeling, goal generation, and dataset reshaping, making it overly complex and heavily reliant on manual engineering design, with limited novelty. In fact, similar ideas have been explored in prior work.
        - Reinformer (ICML 2024 [1]) generates RTG based on state using expectile regression, achieving strong reward stitching capabilities.
        - CoPDT (NeurIPS 2025 [2]) applies a similar concept in offline safe RL, automatically generating RTG distributions conditioned on state and CTG, and sampling from them using expectile regression. It demonstrates strong cost prioritization and achieves safe performance in Safety Gymnasium Navigation tasks under more stringent thresholds ([10, 20, 40, 80]).

        Compared to these works, GAS requires extensive dataset relabeling and augmentation, while methods like Reinformer and CoPDT only rely on learning the expectile distribution, making them simpler and more flexible.

    - The model’s input design—comprising four types of tokens (reward goal, cost goal, RTG, and CTG)—appears unnecessarily complicated. Since the reward goal is already generated, why are RTG and CTG still needed as additional inputs? Although the paper states that reward goal and cost goal are designed to avoid conflicts, what happens when RTG and CTG themselves conflict? How does the model ensure that reward goal and cost goal remain the primary optimization targets, rather than being overshadowed by RTG and CTG? Furthermore, if RTG and CTG do conflict, how is CTG’s priority guaranteed?
- Experiments
    - The experiments are mainly conducted on BulletSafetyGym and the Circle tasks from Safety Gymnasium, but they omit more standard and safety-critical environments, such as other Navigation and Velocity tasks from Safety Gymnasium.
    - The safety requirements in the experimental settings are too lenient.
        - In BulletSafetyGym, satisfying safety constraints is already relatively easy.
        - In Safety Gymnasium, most tasks have constraint thresholds above 20 (for a 10% constraint ratio), which are too loose to evaluate the model’s ability under strict safety conditions.
        - In the “loose constraint” settings, the thresholds remain excessively high, making the evaluation of safety performance insufficient.

        Overall, the experiments mainly verify that GAS achieves high reward under low safety requirements, but its applicability to high-safety-demand scenarios remains questionable.

    - The method involves numerous hyperparameters, including expectile level $\alpha$, reward relabel level $\delta$, dataset reshape threshold q% and sample probability $\epsilon$, among others. The authors should conduct a sensitivity analysis to evaluate how these hyperparameters affect performance.

[1] Reinformer: Max-Return Sequence Modeling for offline RL

[2] Adaptable Safe Policy Learning from Multi-task Data with Constraint Prioritized Decision Transformer

**Questions:**

No other questions.

---

> ### Author Response · Authors · 2025-11-22
>
> We are deeply grateful to the reviewer for the appreciation of our paper, especially on "reasonble approach", "well motivated", and "strong performance".
>
> ### **Q1: The proposed method combines return augmentation, return relabeling, goal generation, and dataset reshaping, making it overly complex and heavily reliant on manual engineering design, with limited novelty.**
> First, most design of GAS are not manual engineering design but theoretical design with convergence and performance improvement guarantee. **To clarify, GAS is not just a special kind of CDT, but is designed purely on stitching dataset where these methods are also theoretical inspired design under this object** (Please look at the next question to understand how GAS differs from other DT-related methods). In Appendix B.2 we theoretically analyze how each part can work with each other and how to improve the theoretical performance on convergence.
>
> Second, Temporal Segmented Return Augmentation and Transition-level Return Relabeling are general to all case where Goal-conditioned GM methods, such as CDT can be applied.  As these method only require signal $(R,C,t)$. If these signals are missing, all GM method, such as CDT and CoPDT, cannot be applied. Otherwise, these two methods can be used normally with theoretical guarantees.
>
> The only engineering design is dataset reshaping. This method is designed to address the "imbalanced hebavior policy" problem, which is already verified in OASIS. In this paper, we just find this problem also exists for GM methods and incorporate this idea to mitigate this problem with proper citiation.

---

> ### Author Response · Authors · 2025-11-22
>
> ### **Q2: Distinguish GAS with Reinformer (ICML 2024 [1]) and CoPDT (NeurIPS 2025 [2]).**
>
> Thank you for improving our related work. We have include the comparison into the end of the Related Work section.
>
> Here are the detailed differences:
>
> **First**, GAS is **NOT EVEN** a Transformer Architecture.  In motivation section, we find that existing problems (like weak stitch and inadequate reward–cost balancing) because the attention block with segment length in CDT cannot capture the relation among transitions correctly. Thus GAS proposes to give up the attention block and segment length to purely focus on stitching dataset. **Thus GAS is purely designed in MLP and GAS's superior performance completely comes from the key contribution of this paper.**
>
> **Second**, Reinformer cannot be applied from Offline RL to Offline Safe RL directly. Following Reinformer, the idea should be expectile regression on RTG and CTG separately and independently to obtain the reward and cost goal function, $R^\pi$ and $C^{\pi'}$, and guide the policy by weight sum of Advantage-Weighted Regression (AWR).
>
> $L^\pi = \beta*AWR(R^\pi, \pi)+(1-\beta)*AWR(C^{\pi'}, \pi)$
>
> However, this way is completely wrong because  $R^\pi$ and $C^{\pi'}$ follows different policy.  $R^\pi$ follows the policy to maximize the reward and $C^{\pi'}$ follows the policy to minimize the cost. Simply combination of them can not provide any theoretically guarantee.
>
> To address this problem, we design the reward and cost goal functions in the way of Eq. (9-12), where both goal functions are jointly optimized to make sure that they are under the same optimization framework (same policy). Eq. (9-12) are not manually designed by derived from the same optimization problem with theoretically guarantee on convergence and performance improvement. **Eq. (9-12) is not a simple utilization of expectile regression but just borrows the key idea of it and designed with carefully theoretical derivation.**
>
> **Third**, CoPDT is a concurrent work (published just 2 months ago) compared with GAS. We believe CoPDT also noticed the problem mentioned above on Reinformer. Thus CoPDT decide to transfer reward goal function $R(|C)$ as cost goal condition. In this way, CoPDT successfuly integrate $R$ and $C$ under the same framework since $R$ is generated by $C$. At this point, we can also find that GAS and CoPDT follows completely different technical route and key idea. The similarity of GAS and CoPDT lies in their objective to solve the same problem: improve stitching ability in OSRL.  Just like the reivewer said, Reinformer and CoPDT using expetile regression. However, GAS rederive the expectile regression to obtain a new update way rather than a standard expectile regression, which just draws on the core idea of expectile regression.
>
> **Fourth**, different from all previous works, GAS is a method that completely focus on stitching dataset. The only thing that GAS does is to seek if there are better transitions with higher rewards and lower costs with the same state. The reason of proposing temporal segmented return augmentation, transition-level return relabeling, and dataset reshaping, is to coordinate with this stitching ability. This is a completely different idea from all previous methods.
>
> As for expectile regression, it is just a tool to approximate maximium value or minimium value within the dataset. It is difficult to say that methods with expectile regression are all similar idea not to mention that Eq.(9-12) is not even a standard expectile regression.
>
> ### **Q3: The model’s input design—comprising four types of tokens (reward goal, cost goal, RTG, and CTG)—appears unnecessarily complicated. Since the reward goal is already generated, why are RTG and CTG still needed as additional inputs?**
>
> We agree that provide reward goal and cost goal can tell the policy what the ideal values are. However, when RTG and CTG are also provided, the model can also extract the co-feature of (reward goal $\leftrightarrow$ RTG) and (cost goal $\leftrightarrow$ CTG) in the neural network. This design can give the model additional information on "How RTG is transfered to reward goal", "How CTG is transfered to cost goal", "How RTG is distilled to reward goal", and "How CTG is distilled to cost goal" in the training procedure.

---

> ### Author Response · Authors · 2025-11-22
>
> ### **Q4: Although the paper states that reward goal and cost goal are designed to avoid conflicts, what happens when RTG and CTG themselves conflict?**
>
> If RTG and CTG conflicts, the model will always ensure CTG's priority guarantee.
>
> Considering the return relabeling method alone, it work like "randomly specify targets" as the input of GAS, making it robust enough to different manually specified targets during test time. Combining it with the goal function, the goal function takes **relabeled targets as input** but still **expectile regression on the true/real returns**. In this way, the goal function can give correct maximum goal given any "randomly specify targets" and thus is robust in test stage.
>
> For example, in the training, assume the RTG and CTG in the transition is $(R,C)$, following the random return relabeling methods, we get $(\hat R,\hat C>C)$, where the values are different for each relabeling following Eq. (8).
>
> $R$ and $C$ comes from the dataset, which has no conflicts. $\hat R$ and $\hat C$ are randomly generated and may conflict.
>
> Then the goal function take $(\hat R,\hat C)$ as input but the loss is calculated on $(R,C)$. The goal function $V^R$ and $V^C$ works like:
>
> $V^R(\hat R,\hat C) = $Eq.(9-10) on $R$ , **the equation is constrained by** $\hat C$, **to ensure CTG's priority guarantee.**
>
> $V^C(\hat R,\hat C) = $Eq.(11-12) on $C$ , **Eq.(12) is controled by Eq. (9), where only safe CTG can be updated with high weight.**
>
> In this way, the goal function learns "How CTG/RTG is related to cost/reward goal" and works like a function to transfer conflict pairs to non-conflicting output, similar to Q3.
>
> During the test stage, the manually defined RTG and CTG is similar to $\hat R$ and $\hat C$. However, during the training stage, the goal function already learned how to transfer them to non-conflicting outputs.
>
> ### **Q5: The experiments are mainly conducted on BulletSafetyGym and the Circle tasks from Safety Gymnasium, but they omit more standard and safety-critical environments, such as other Navigation and Velocity tasks from Safety Gymnasium.**
> Thank you for your suggestion. We have add 5 additional Velocity tasks in SafetyGym into comparison in Table 5, Appendix E.3. Compared with two newest and solid baselines (CAPS and CCAC) and another GM-assisted baseline (CDT), GAS successfuly achieves safe and best performance in most tasks. In particular, GAS performs better reward maximization ability on "loose constraint" and better constraint satisfaction ability on "tight constraint". These results are similar to other tasks, which demonstrate GAS's superiority on adressing the proposed problems.
>
> ### **Q6: In BulletSafetyGym, satisfying safety constraints is already relatively easy.**
> We appreciate the reviewer's suggestion to improve our experiment settings and tasks. We would like to clarify the difficulty on BulleySafetyGym and Safety Gymnasium. For CDT related methods, it remains unclear that the Velocity tasks in Safety Gymnasium are more complex than the tasks in BulletSafetyGym.
>
> **The challenges faced by BulletSafetyGym are different from the Safety Gymnasium. In particular, the challenges of Velocity tasks are "longer trajectory" and "higher cost".** This is indeed a difficult challenge for RL-base method but CDT is naturally suitable for these tasks.
>
> Looking at the dataset visualization provided by DSRL, we can find that the dataset distribution in Velocity is similar to that in CarRun of BulletSafetyGym. Their common feature is that the reward returns and cost returns are independent when rewards are maximized. In particular, in these environments, to achieve $R^{max}$, the cost returns can be ranged from $0$ to $C^{max}$.
>
> Since CDT related methods are goal conditioned BC, it is easy to choose feasible $R$ and $C$ without conflicts. From the perspective of CDT, the Velocity task is not much more difficult than CarRun. This claim can also be validated by the experiment results where the performance of CDT on CarRun is the same as performance on Velocity tasks.
>
> **However, most tasks in BulletSafetyGym follows a strict proportional relationship on $R$ and $C$, where $R$ increases as $C$ increases. This is the main challenge in BulleySafetyGym where CDT shows disadvantages.** In this case, CDT related methods are more easily to choose infeasible $(R,C)$ pairs that may conflicts with each other. **That is why CDT can achieve optimal and safe performance on Velocity tasks easily while fails to be safe in some tasks of BulletSafetyGym, no matter in this paper or previous works of baselines.**
>
> Overall, due to different features of two benchmarks, we test our method on both benchmarks to comprehensively evaluate our method. Besides, the "tight" constraint in BulleySafetyGym are much smaller to improve the difficulty where CDT fails while GAS shows good performance. Thus these results fairly demonstrate GAS's superiority against baselines.

---

> ### Author Response · Authors · 2025-11-22
>
> ### **Q7: The safety requirements in the experimental settings are too lenient.**
> We respectfully disagree with reviewer on this point. The requirements are selected with careful consideration to ensure a standard and fair comparison.
>
> - In BulletSafetyGym, most constraints in this paper are tighter than existing papers. We have explain in Q6 that BulleySafetyGym is not a simple benchmark for DT-related methods.
> - In BulletSafetyGym, CPQ, COptiDICE, VOCE and CDT fails to satisfy the constraint in most tasks, CAPS and CCAC fails in some tasks, while WSAC and FISOR are overly conservative even though safe. This result shows that balance reward maximization and constraint satisfaction is not easy.
> - In Safety Gymnasium (PC1, PC2, CC1,CC2), CPQ, COptiDICE, WSAC, VOCE, CDT, FISOR, and CCAC fails to satisfy the constraint while CAPS only satisfy the constraint on two tasks. This result already shows that the "tight constraint" is tight enough since almost all existing baselines fail to be safe. However, GAS achieves safe performance on these tasks.
> - In OSRL, there is no widely accepted definition on "tight" or "loose" constraint. We made fair comparison under reasonable settings and show that GAS can improve CDT on reward maximization and constraint satisfaction. As for CoPDT, it is a concurrent work compared with GAS. Although CoPDT chooses smaller constraints than ours, it is hard to say that CoPDT is tight while GAS is not tight.
> - In the “loose constraint” settings, most methods fail to maximize the reward compared to GAS and some methods even performs unsafely. If the thresholds remain excessively high, all baselines should achieve maximized reward within the constraint.
> - As we explained in the paper, "tight constraint" focuses on evaluation of safety performance, where most algorithm fails to be safe, and ''loose constraint" tests the reward maximization ability. Based on this purpose, only GAS achieves maximized and safe performance on most tasks.
>
> ### **Q8: The method involves numerous hyperparameters, including expectile level $\alpha$, reward relabel level $\delta$, dataset reshape threshold q% and sample probability $\epsilon$, among others. The authors should conduct a sensitivity analysis to evaluate how these hyperparameters affect performance.**
>
> Thank you for your suggestion. We have added the analysis on expectile level in Figure. 7, Section 6.4, and reward relabel level in Figure. 6, Section 6.3. The hyper parameter analysis is shown in Figure. 11 and Figure. 12 in Appendix E.5.
>
> Here is the result in summary:
> - GAS is robust to a wide range of $\alpha$ as long as $\alpha$ is not to large ($\alpha=0.99$) or to small ($\alpha=0.5$).
> - As reward relabel level $\delta$ increases, the robustness of GAS on infeasible targets also increases. Besides, GAS is also robust to $\delta$ due to the optimization based on expectile regression.
> - Sample probability and Dataset reshape threshold on dataset reshaping mainly influence the training efficiency of GAS. The final performance can be influenced only when the parameters are close to GAS w/o DR.

---

> > ### Comment · Reviewer_aKUM · 2025-11-25
> >
> > Thank you very much for the authors’ detailed responses. These replies have indeed addressed some of my concerns, but several concerns still remain.
> >
> > 1. **Methodology** — I accept that GAS is a method specifically designed for stitching datasets, and that Temporal Segmented Return Augmentation and Transition-level Return Relabeling are two techniques that can be flexibly applied to goal-conditioned GM methods; these are indeed meaningful contributions of GAS. I also accept that Reinformer cannot be used for offline safe RL and that CoPDT is concurrent work.
> >
> >     My remaining concern lies in the extrapolation error caused by potential conflicts in the policy inputs $(\hat{R},\hat{C},V_R,V_C)$, specifically the conflict between $(\hat{R},\hat{C})$. Of course, during GAS training, neither $(V_R,V_C)$ nor $(\hat{R},\hat{C})$ will conflict, since they are sampled from the real offline dataset. However, conflicts can arise during policy deployment, as the authors themselves noted in the rebuttal, due to manually defined RTG and CTG.
> >
> >     For example, suppose we desire both very high reward and very low cost at test time, so we set RTG = 100 and CTG = 10. Yet in the training data, the highest RTG paired with CTG = 10 might be only 40, while the lowest CTG paired with RTG = 100 might be 30. How can we ensure that the policy prioritizes safety in such a case? From the perspective of offline RL, such (RTG, CTG) pairs are clearly OOD. Even if we assume the goal function $(V_R,C_V)$ can produce reasonable goals, the input $(\hat{R},\hat{C})$ is already OOD—let alone the fact that $(\hat{R},\hat{C})$ also feeds into $(V_R,C_V)$, generating further OOD issues. Therefore, I believe that using $(\hat{R},\hat{C},V_R,V_C)$ as policy inputs may still lead to reward–cost conflicts at test time.
> >
> > 2. **Experiments** — I appreciate the authors’ effort to conduct experiments on the Velocity tasks and perform a hyperparameter sensitivity analysis. I also agree that GAS exhibits stronger stitching capability than prior methods such as CDT when the safety constraints are not very strict, thereby achieving higher returns.
> >
> >     I am curious, however, about how GAS would perform under stricter constraints (10 or 20) in Safety Gymnasium’s navigation tasks (CarButton1–2, CarPush1–2, CarGoal1–2, PointButton1–2, PointPush1–2, PointGoal1–2), especially in comparison with CDT, CAPS (IQL), and CCAC.
> >
> >
> > Taking all the responses into account, I am inclined to keep my score.

---

> > > ### Author Response · Authors · 2025-11-27
> > >
> > > Thank you for your timely response and follow-up questions. We are very delighted to resolve some of your concerns. We would like to clarify the remaining points as follows.
> > >
> > > ### **Q1: How does GAS policy prioritize safety in such a case?**
> > >
> > > The key component to deal with the reviewer's concerns lies in the combination of return relabeling.
> > >
> > > Just follow the reviewer's example (RTG=100 and CTG = 10), where (RTG=100 and CTG=(30$\to$40)) in the dataset.
> > >
> > > **First**, based on Eq. (8), (RTG=100 and CTG=(30$\to$40)) will not be utilized for the training of CTG=10 because CTG=(30$\to$40) can only be relabeled to be larger.
> > >
> > > **Second**, let's also assume the dataset, whose CTG $\leq 10$ follows (RTG=$X$, CTG$\leq 10$). This assumption is natural since we should have the data with CTG $\leq10$. Based on Eq. (8), these data with CTG $\leq10$ is possible to be relabeled to CTG $=10$, with (RTG=$X\pm\delta X$, CTG$= 10$), and the relabeled data will be trained under Eq. (9-13). Therefore, as long as RTG=$X\pm\delta X$ can be 100, then GAS's goal functions will project (RTG=100 and CTG = 10) back to (RTG=X and CTG $\leq$ 10) to prioritize safety in such a case and then find the better value under expectile regression. This is why the ability to prioritizes safety increases  .
> > >
> > > **Third,** this method indeed improves the ability to prioritizes safety as shown in Figure 6, the hyper-parameter analysis of reward relabel level $\delta$:
> > >
> > > **(1)** As $\delta$ increases, GAS becomes much more robust to deal with different RTG when CTG is not changed.
> > >
> > > **(2) In BallRun task, the dataset follows (RTG $\leq430$, CTG=16) while GAS is robust for (RTG=500, CTG=16), where the dataset with RTG=500 follows (RTG=500, CTG=(25$\to$ 80)). We think this case perfectly fits with the example provided by the reviewer, and GAS successfully prioritized safety first.**
> > >
> > > **Finally**, we agree that this relabel method is not suitable for pretty extreme examples. For example, users give (RTG=900, CTG=10) where the dataset follows (RTG=80, CTG$\leq$10). This extreme example is nearly impossible and unreasonable to happen since the user should also know that this extreme case is impossible to achieve, considering the dataset information.
> > >
> > > ### **Q2: GAS performance for constraint 10 or 20 on other navigation tasks.**
> > >
> > > We are continuously supplementing the experimental results according to your requirements and will present the results in the next few days.

---

> > > > ### Author Response · Authors · 2025-12-01
> > > >
> > > > ### **Q2: GAS performance for constraint 10 or 20 on other navigation tasks.**
> > > >
> > > > Dear reviewer, we have supplemented the experiments based on your requirement (constraint=10 or 20 in navigation tasks).
> > > >
> > > > 1. As shown in the following table, GAS still achieves safe results on all tasks and best performances on most tasks for both thresholds = 10 and 20 compared with other baselines. **This result is similar to our main result in Table 1 of this paper.**
> > > > 2. Compared with CDT, our methods show significant advantages in "stitching" ability to achieve much safer results. **This result also successfully validates how GAS addresses CDT's weakness in the Motivation section.**
> > > > 3. Compared with CAPS, GAS can achieve higher rewards in most cases. **Besides, as shown in Table 1 of this paper, GAS performs better than CAPS on most other tasks.**
> > > >
> > > > Overall, we have compared 8 latest baselines on 21 tasks (8 for BulletGym, 5 for Velocity tasks, and 8 for navigation tasks) for a wide range of thresholds. **All experiment results can demonstrate that GAS significantly improves the stitching ability compared with existing GM-assisted methods and successfully prioritizes cost targets over reward targets.**
> > > >
> > > > | Task             | **CDT** |      | **CAPS**  |          | **CCAC** |          | **GAS**   |          |
> > > > | ---------------- | ------- | ---- | --------- | -------- | -------- | -------- | --------- | -------- |
> > > > | **Threshold=10** | R       | C    | R         | C        | R        | C        | R         | C        |
> > > > | PointPush1       | 3.09    | 32.3 | **1.85**  | **9.5**  | 0.99     | 11.8     | **1.82**  | **7.29** |
> > > > | CarPush1         | 4.11    | 26.6 | **2.41**  | **5.09** | **-4.7** | **6.3**  | **3.2**   | **8.0**  |
> > > > | PointGoal1       | 20.89   | 23.0 | **8.67**  | **9.73** | 22.22    | 42.7     | **10.41** | **9.0**  |
> > > > | CarGoal1         | 25.67   | 38.0 | **11.66** | **8.69** | 31.45    | 62.2     | **10.26** | **7.8**  |
> > > > |                  | **CDT** |      | **CAPS**  |          | **CCAC** |          | **GAS**   |          |
> > > > | **Threshold=20** | R       | C    | R         | C        | R        | C        | R         | C        |
> > > > | PointPush1       | 4.11    | 27.3 | **2.06**  | **13.8** | **2.07** | **13.9** | **2.44**  | **12.9** |
> > > > | CarPush1         | 5.09    | 27.3 | **3.35**  | **13.3** | **-3.5** | **5.4**  | **3.4**   | **14.5** |
> > > > | PointGoal1       | 21.5    | 38.2 | **12.63** | **15.5** | 22.63    | 45.9     | **13.53** | **17.8** |
> > > > | CarGoal1         | 26.38   | 43.6 | **10.3**  | **13.2** | 32.08    | 66.59    | **16.57** | **15.6** |

---

### Official Review · Reviewer_oGCt · 2025-10-31

**Soundness:** 3
**Presentation:** 3
**Contribution:** 3
**Rating:** 6
**Confidence:** 4

**Summary:**

This paper starts from two challenges in offline safe RL: 1) most existing methods lack capability to stitch good sub-trajectories from different sub-optimal trajectories, and 2) they struggle to balance reward and cost. The authors propose Goal-Assisted Stitching (GAS), which includes return augmentation and relabeling, expectile regression, dataset distribution reshaping, to overcome those issues. The authors run experiments on safe RL benchmarks and the results show that GAS show superior performance to baselines.

**Strengths:**

- Two motivations of paper, i.e., lack of stitching and failure to balance reward and cost, are clearly presented and important to safe offline RL community.
- GAS exhibits consistent advantage over most baselines on DSRL in terms of reward improvement and cost constraint satisfaction, in both situations with tight or loose constraint.
- The experiment is comprehensive. The authors also provide ablations to explain the motivation or show the effectiveness of relabel module.

**Weaknesses:**

- The "stitching" ability of the GAS is a little over-claimed. The authors claim the GAS has the capability to stitch the sub-trajectories. However, GAS seems to learn new return-to-go and cost-to-go targets instead of stitching the data. See more in the question section. I understand that the true stitching can be hard but I believe there is a large mismatch between "stitching" and GAS's implementation.

**Questions:**

- In fig 2, the stitched trajectory (denoted by red arrow) is confusing. For example, in the stitch of GAS showed by the second arrow (from the 2nd block of $\tau_j$ to the 1st of $\tau_k$), since the states in $\tau_j,\tau_k$ will be different, how do you make sure the 1st block of $\tau_k$ can be stitched after the 2nd block of $\tau_j$? Otherwise it's not even a valid trajectory.
- In sec.5.2, GAS seeks better $R,C$ based on other sub-trajectory / segmentation as long as their lengths are the same. However, since the starting states $s_t, s_{t'}$ are different, their optimal $R$ and $C$ will also be different. Why do you think the $R,C$ of $s_{t'}$ can be a good reference for $s_t$?
- The dataset reshaping seems to be also an important component. Do you have any ablation on that? E.g., the performance comparison with v.s. without reshaping.

---

> ### Author Response · Authors · 2025-11-22
>
> We are delighted by the reviewer's praise for our article, especially on (1) important motivation, (2) clear representation, and (3) comprehensive experiment.
>
> ### **Q1: The "stitching" ability of the GAS is a little over-claimed. The authors claim the GAS has the capability to stitch the sub-trajectories. However, GAS seems to learn new return-to-go and cost-to-go targets instead of stitching the data. See more in the question section. I believe there is a large mismatch between "stitching" and GAS's implementation.**
>
> Thank you for your correction. We have clarify the difference of "stitching" ability in RL and GM methods, in **GM Methods in Offline RL** part of Related Work section in the updated manuscript.
>
> In optimization, the stitching ability is to find optimal policy from sub-optimal data. In this paper, "stitching" is compared with other GM-assisted method like CDT, which is purely goal conditioned behavior cloning. CDT related methods fit the training data by conditioning on the reward-to-go at the trajectory level to avoid sampling OOD actions during the training phase, resulting in a stable, general, and high-performing framework. However, behavior cloning methods inherently lacks the ability to perform better beyond the dataset trajectory, which is called "lack of stitching ability".
>
> **Therefore, in goal-conditioned framework, the "stitching" ability indicates "the ability to find a better policy beyond the dataset trajectory (goal-conditioned BC)".** Most Offline RL based on DT, like EDT [1], ADT [2], and Reinformer [3] focus on the latter understanding on "stitch". Therefore, the assertion that "stitching is a crucial ability that GM lacks" has already been discussed in previous works on Offline RL.
>
> Following this understanding on "stitching" ability, GAS is proposed to **improve the stitching ability** of GM-assisted method in OSRL.  Different from CDT, at each step, GAS tries to seek transition, whose action can achieve high reward goal within the constraint for the same state, which looks like "stitching" rather than cloning. Although GAS does not explicitly stitching different trajectories into new trajectories in the dataset, the way on how GAS works can be regarded as a kind of "implicit stitching" via continually seeks better transitions beyond the dataset trajectory.
>
> ### **Q2: About Figure 2**
>
> Sorry for the confusion. We certainly agree with you that only the same state can be stitched. We have improve the Figure 2 to improve clarity.
>
> To clearly distinguish transition transfer and stitching, we now use **red solid line** to show that the transition is acutally transferred to next state and **red dashed line** to show that it is just a stitching procedure under the same state. The red solid line only connect from good transition to good transition (blue color) or to common transitions (orange color). Once GAS meets common transitions, it start to seek good transitions, whose action can achieve high reward goal within the constraint for the same state. The stitching procedure (red dashed line) is not "transition transfer".
>
> ### **Q3: About section 5.2**
>
> We agree with you that only same state $s_t=s_{t'}$ can be stitched. We did not emphasize it because it is impossible to regard two different states as the same one. We have include this emphasis into Section 5.2 in the updated manuscript.
>
> Besides, here is an exact example to understand Temporal Segmented Return Augmentation (TSRA) if you have further concerns on section 5.2.
>
> **Example:**
>
> For a state at time $200$, whose reward goal and cost goal are considered from time $200$ to $300$, there may existing the same state but better reward goal and cost goal considered from time $100$ to $200$ in another trajectory.
>
> **More detail:**
>
> For a state $s_{200}=s$, $R_{200:300}$ and $C_{200:300}$ are the cumulative values from $200$ to $300$. This state may also shown in other trajectories at different time step, like $(s_{100}'=s,R_{100:300}', C_{100:300}')$. Without TSRA, these transition will be regarded as different transitions due to time embedding. However, we know $(s_{100}'=s,R_{100:300}', C_{100:300}')$ can be augmented to $(s_{100}'=s,R_{100:200}', C_{100:200}')$ if the last 100 steps are not considered. Since both $R_{200:300}$ and $R_{100:200}'$ considers future 100 steps, we can regard $(s_{100}'=s,R_{100:200}', C_{100:200}')$ as a better transition if $R_{100:200}'>R_{200:300}$ and $C_{100:200}'\leq C_{200:300}$. Because the state considers from $200$ to $300$, we need to align this time length, which is the description "as long as $\Gamma-t=T-t'$ "

---

> ### Author Response · Authors · 2025-11-22
>
> ### **Q4: The dataset reshaping seems to be also an important component. Do you have any ablation on that? E.g., the performance comparison with v.s. without reshaping.**
>
> The ablation study on dataset reshaping, including detailed hyperparameter analysis is provided in Appendix E.5.  Experiment results demonstrate that GAS is robust under different $q\%$ and $\epsilon$ of dataset reshaping.
>
> Besides, we also provide detailed hyper-parameter analysis, including, expectile level $\alpha$ to Section 6.4 and Figure 7, reward relabel level in Figure 6, and dataset reshaping in Figure. 11 and 12, Appendix E.5 in the updated manuscript if you have further concerns on experiments.

---

> ### Author Response · Authors · 2025-11-28
>
> Dear reviewer, as for your first question about the stitching ability, we also have included this as a limitation of the Goal-conditioned framework, as well as GAS, in the Conclusion Section:
>
> "Besides, due to the limitation of the existing goal-conditioned framework, GAS's stitching ability focuses on improving performance by better selecting high-quality sub-segments in the safe dataset, rather than constructing new safe trajectories by combining safe and unsafe data in RL."
>
> Finally, we still want to clarify that **this limitation comes from the basic structure of GM-assisted methods rather than GAS**. GM assisted methods give up the "Bellman backup stitch" ability for zero shot adaptation ability and improved stability. **Nevertheless, GAS still significantly improves the stitching ability of the standard GM-assisted methods as shown in the experiment results.**

---

### Official Review · Reviewer_RPXE · 2025-11-01

**Soundness:** 3
**Presentation:** 3
**Contribution:** 3
**Rating:** 4
**Confidence:** 3

**Summary:**

The paper introduces a method called Goal-Assisted Stitching (GAS) to improve how reinforcement learning agents make safe and effective decisions using only offline data. It builds on recent generative model approaches but points out that these methods often fail to combine good parts of different trajectories and cannot properly balance rewards and safety limits. GAS addresses these issues by estimating achievable reward and cost goals directly from the dataset, instead of depending on user-specified targets. It does this through a statistical technique called expectile regression and adds two simple but effective steps: segmenting trajectories into shorter pieces to create more training samples, and reshaping the dataset so that the model sees a more even mix of safe and risky examples. Tested on common safe RL benchmarks, GAS shows more reliable safety under strict constraints and higher returns when the safety limits are relaxed, performing better than previous approaches in both stability and adaptability.

**Strengths:**

- This work tackles a practical and important problem in Offline Safe Reinforcement Learning (OSRL) with Generative Models (GMs), focusing on two key challenges: balancing reward and cost, and improving the model’s ability to stitch useful transitions from different trajectories.
- The paper presents solid theoretical support for the proposed GAS algorithm, especially through its use of expectile regression to estimate optimal reward and cost goals without relying on Bellman backups. This design effectively avoids common offline RL problems like value overestimation and out-of-distribution action selection. The clear theoretical foundation not only strengthens the methodological soundness of the approach but also enhances confidence in the reported empirical results, marking it as a notable strength of the work.
- The paper shows strong empirical rigor through comprehensive experiments on multiple benchmarks and baselines under various constraint settings and ablation studies, convincingly demonstrating the robustness and effectiveness of the proposed GAS method.
- The paper is clearly written and well organized, moving smoothly from the motivation to the method and then to the experiments, making it easy for readers to follow the main ideas and technical details.

**Weaknesses:**

- The reward and cost goal functions are learned from offline data using expectile regression, which makes them susceptible to the biases of the dataset. If the data are unbalanced or lack high-quality transitions, the estimated goals can become either too optimistic or too conservative, leading the policy in the wrong direction. Because these functions guide the policy’s optimization, even small estimation errors can distort the balance between reward and cost or result in unsafe actions.
- The use of expectile regression introduces a hyperparameter \alpha that controls the aggressiveness of the upper-tail estimation. The paper does not analyze the sensitivity of \alpha or provide adaptive tuning [1].
- Although the paper includes some information on infrastructure and training time, it does not clearly analyze computational efficiency. The method adds several components—segmented augmentation, dual goal functions, and dataset reshaping—but lacks a direct runtime or scalability comparison with CDT, making it difficult to assess the true efficiency impact [2].
- While the proposed GAS framework demonstrates strong performance on dense-reward benchmarks such as SafetyGymnasium and Bullet-Safety-Gym, it lacks evaluation in sparse-reward settings [3-5].
- Although the paper builds on recent progress in generative model–assisted offline safe RL, earlier works [6] have already explored similar ideas such as goal-conditioned augmentation and return relabeling. It would strengthen the paper to more clearly distinguish GAS from these methods, include [6] as a baseline for direct comparison, and incorporate additional recent strong baselines to ensure a fair and comprehensive evaluation [7-8].
- The reward and cost goal functions are trained separately, without accounting for their inherent correlation in constrained settings. This independent training can produce inconsistent or even infeasible goal pairs, but the paper does not discuss or analyze this potential issue.

**Questions:**

- How sensitive is the performance of GAS to the choice of the expectile level \alpha?
- What are the training time and GPU resource requirements of GAS compared with CDT?
- How would GAS perform on tasks with sparse or delayed rewards?
- Since the reward and cost goal functions are trained independently, could the authors analyze their mutual consistency and verify that the resulting goal pairs correspond to feasible points on the reward–cost frontier?
- Could the authors include additional baselines, as mentioned in the weakness section, to further verify the performance of GAS?

## Reference

**[1]** Zhuang, Zifeng, et al. "Reinformer: Max-return sequence modeling for offline rl." arXiv preprint arXiv:2405.08740 (2024).

**[2]** Chemingui, Yassine, et al. "Constraint-adaptive policy switching for offline safe reinforcement learning." Proceedings of the AAAI Conference on Artificial Intelligence. Vol. 39. No. 15. 2025.

**[3]** Cen, Zhepeng, et al. "Learning from sparse offline datasets via conservative density estimation." arXiv preprint arXiv:2401.08819 (2024).

**[4]** Rengarajan, Desik, et al. "Reinforcement learning with sparse rewards using guidance from offline demonstration." arXiv preprint arXiv:2202.04628 (2022).

**[5]** Yamagata, Taku, Ahmed Khalil, and Raul Santos-Rodriguez. "Q-learning decision transformer: Leveraging dynamic programming for conditional sequence modelling in offline rl." International Conference on Machine Learning. PMLR, 2023.

**[6]** Wang, Ruhan, and Dongruo Zhou. "Safe Decision Transformer with Learning-based Constraints." 7th Annual Learning for Dynamics\& Control Conference. PMLR, 2025.

**[7]** Guan, Jiayi, et al. "Voce: Variational optimization with conservative estimation for offline safe reinforcement learning." Advances in Neural Information Processing Systems 36 (2023): 33758-33780.

**[8]** Wei, Honghao, et al. "Adversarially trained weighted actor-critic for safe offline reinforcement learning." Advances in Neural Information Processing Systems 37 (2024): 52806-52835.

---

> ### Author Response · Authors · 2025-11-22
>
> We are delighted by the recognition of our contribution from the reviewer, especially on which (1) the problem this work tackling is parctical and important, (2) solid theoretical support, strong empirical rigor and comprehensive experiments, (3) clearly written and well organized. Regarding reviewer's concerns, we also provide the following clarification:
>
> ### **Q1: Expectile regression makes goal functions susceptible to the biases of the dataset, especially for unbalanced or lack high-quality transitions.**
>
> Thank you for your concerns. This problem belongs to the issue with goal-conditioned BC for all GM methods in OSRL, but not for expectile regression in GAS. The goal-conditioned framework make the policy susceptible to the influence of unbalanced or low-quality transitions due to imbalanced reward and cost targets. In contrast, the goal functions learned through expectile regression help mitigate these biases rather than exacerbate them.
>
> **Unbalanced Data:**
>
> For constraint $C$, CDT is trained by cloning the data behavior with $(R,C)$, which can be influenced to be unsafe if the data around $(R,C)$ is unbalanced. However, as shown in theoretical analysis, GAS considers all data from $(R,0)$ to $(R,C)$ due to relabeling and expectile regression for constraint $C$. This indicates that the expectile regression will enforce GAS to learn a comprehensive value from data in the interval $(R,0\to C)$. Learning from $(R,0\to C)$ should be much more balanced than learning solely from $(R,C)$, thereby allowing expectile regression to effectively mitigate biases within the dataset..
>
> **Low-quality transitions**
>
> In scenarios where all trajectories in the dataset exhibit low quality (e.g., they have low reward targets), the reward and cost targets during the testing phase may be completely out-of-distribution (OOD) compared to the training data. For example, the training data involves $R\in(0,500)$ while the testing targets is $550$.
>
> In this case, CDT can be easily influenced since $550$ is not shown in the training stage. There is no guarantee on the performance of CDT in such case. In contrast, GAS is also proved theoretically to consider cost with priority and project $550$ back to the dataset distribution due to return relabeling.
>
> ### **Q2: The use of expectile regression introduces a hyperparameter $\alpha$ that controls the aggressiveness of the upper-tail estimation. The paper does not analyze the sensitivity of $\alpha$ or provide adaptive tuning [1].**
>
> Thank you for improving our experiment. We have include the hyperparameter $\alpha$ to Section 6.4 and Figure 7 in the updated manuscript. The experiment results shows that GAS is robust to a wide range of $\alpha$, as long as $\alpha$ is not too large.
>
> Besides, we also analyze other hyper-parameter, including reward relabel level in Figure 6, and dataset reshaping in Figure. 11 and 12, Appendix E.5 in the updated manuscript.
>
> ### **Q3: Although the paper includes some information on infrastructure and training time, it does not clearly analyze computational efficiency. The method adds several components—segmented augmentation, dual goal functions, and dataset reshaping—but lacks a direct runtime or scalability comparison with CDT, making it difficult to assess the true efficiency impact [2].**
> The extra overhead is very small, including TSRA, dual goal function, and dataset reshaping. Here is detailed explanation:
>
> TSRA and Dataset reshape is a rule-based dataset augmentation method with a negligible computational cost, typically requiring only extra few minutes compared over the overall training time. Importantly, TSRA and Dataset reshape **just need to be executed once** when processing the dataset before training models, and the augmented dataset can be stored for future use without incurring additional computational costs.
>
> Besides, GAS is not a Transformer Architecture but MLPs. Compared with CDT, the only additional computational costs come from training two additional dual goal functions, which are also MLPs. In motivation section, we find that existing problems (like weak stitch and inadequate reward–cost balancing) because the attention block with segment length in CDT cannot capture the relation among transitions correctly. Thus GAS proposes to give up the attention block and segment length to purely focus on stitching dataset. Therefore, the time complexity of GAS, which is purely designed with MLP, is much more lower than CDT with Transformer Architecture.
>
> Finally, the training efficiency in offline RL is not as important as that in online RL. In online RL, the agent get data by continuously interacting with the environment, which is time consuming. In contrast, offline RL are trained on a pre-collected and fixed dataset. In this case, the training efficiency of offline RL is closer to supervised learning, where the computational efficiency of MLP is higher than that of Transformer.

---

> ### Author Response · Authors · 2025-11-22
>
> ### **Q4: GAS lacks evaluation in sparse-reward/delayed-reward settings [3-5].**
> We really appreciate your suggestion. However, we do not test in a sparse-reward environment **because OSRL lacks such environments and datasets as far as we know**, which may be a future direction for datasets and benchmarks. Unlike the online setting, the environment and dataset used in the offline setting should be well-recognized, as OSRL only has the DSRL library. That why most existing papers design experiments on DSRL.
>
> Suggested references [3-5] that tests sparse-reward settings, like Maze, belong to Offline RL rather than OSRL. **Unfortunately, these environments and datasets in Offline RL can not be used for OSRL since OSRL focus on cost signal while Offline RL only provides reward signal.**
>
> Besides, even without sparse-reward settings, our evaluation is comprehensive and solid to show the improvement of GAS against existing GM-assisted methods like CDT. We evaluate on 2 benchmark, including 17 tasks (8 tasks in Bullet-safety-gym and 9 in Safety-gymnasium) and compared with 8 baselines.
>
> ### **Q5: About CQDT [6].**
> Thank you for making our related work more comprehensive by mentioning CQDT. We have included the disscusion on CQDT into Related Work section. Besides, we also extend the following discussion for a detailed explanation:
>
> **Comparison with CQDT and GAS.**
>
> First, The key contribution of CQDT is relabeling the Reward-to-go and Cost-to-go by training lots of CPQ under different constraints. The problem of this method is shown in the last paragraph of related work. In offline RL, it is acceptable to utilize Q function to guide DT's optimization because Offline RL only has one objective: maximize reward. However, in OSRL, CPQ can only deal with one constraint threshold after training. Since CDT has zero-shot ability and CPQ doesn't. **Ideally, CPQ should be retrained on every possible constraint to improve CDT, which is very inefficient.** The key contribution of GAS, which combines goal function and expectile regression, can improve the stitching ability without any retraining.
>
> **The augmentation method, named in CQDT, is actually relabeling, which is totally different from our augmentation method.** Our Temporal Segmented Return Augmentation (TSRA) is an augmentation on time-step, which really enriches the data beyond the dataset. If we have to say similarity, augmentation in CQDT is a bit like the relabeling method in our paper but with totally different technological routes.
>
> We have included the newest and SOTA baselines like CAPS and CCAC. Besides, we also include lots of recent work from well-known conferences, such as VOCE and WSAC. For some early but solid works, like CPQ and COptiDICE, we also include them in comparison. **We think the baselines in our paper are already comprehensive and convincing.**
>
> From the experimental perspective , CQDT compares with 6 old baselines (before 2023) on 5 tasks while GAS compares with **8 latest baselines (2022-2025) on 17 tasks** to demonstrate the superiority.
>
> Finally, as a concurrent work, CQDT doesn't provide any official code as far as we know so that it is hard to incorporate it into comparison.

---

> ### Author Response · Authors · 2025-11-22
>
> ### **Q6: Incorporate additional recent strong baselines to ensure a fair and comprehensive evaluation [7-8].**
>
> Thank you for your suggestion. VOCE [7] and WSAC [8] are already included in Table 1 of this paper. Additionally we also compares most recent research: CCAC [**A**] (ICLR 2025) and CAPS [**B**] (AAAI 2025) in Table 1 in the updated manuscript. Compares with these baselines, GAS remains competitive on both types of constraints and all tasks.
>
> **[7]** Guan, Jiayi, et al. "Voce: Variational optimization with conservative estimation for offline safe reinforcement learning." Advances in Neural Information Processing Systems 36 (2023): 337C58-33780.
>
> **[8]** Wei, Honghao, et al. "Adversarially trained weighted actor-critic for safe offline reinforcement learning." Advances in Neural Information Processing Systems 37 (2024): 52806-52835.
>
> [**A**] Guo, Z., Zhou, W., Wang, S., & Li, W. (2025, March). Constraint-conditioned actor-critic for offline safe reinforcement learning. In *The Thirteenth International Conference on Learning Representations*.
>
> [**B**] Chemingui, Y., Deshwal, A., Wei, H., Fern, A., & Doppa, J. (2025, April). Constraint-adaptive policy switching for offline safe reinforcement learning. In *Proceedings of the AAAI Conference on Artificial Intelligence* (Vol. 39, No. 15, pp. 15722-15730).
>
> ### **Q7: The reward and cost goal functions are trained separately, without accounting for their inherent correlation in constrained settings. This independent training can produce inconsistent or even infeasible goal pairs, but the paper does not discuss or analyze this potential issue.**
>
> The reward and cost goal function are not trained separately but **under the same optimization framework**. We have clarify it after Eq. (12) in Section 5.4 in the updated manuscript.
>
> In Eq. (9), the advantage function for **reward** goal contains **cost** goal as constraint. Besides, Eq (12) is also not a simple expectile regression. In Eq (12), the advantage of **reward** goal control the expectile value and is multiplied to the square of **cost** goal advantage. This formulation is not manually designed but derived from the same optimization problem.
>
> **In Appendix B.1, we show how to derive the representation and loss function of reward goal function, cost goal function, and policy.** The reward goal function converges to the maximum achievable rewards within the constraint estimated by the cost goal function. The cost goal function estimate what the cost value of the corresponding reward goal function is. Both reward goal function and cost function follows the same policy in Eq. (20). Thus both functions are impossible to produce inconsistent/infeasible goal pairs with theoretical guarantee.

---

### Official Review · Reviewer_gTZL · 2025-11-01

**Soundness:** 2
**Presentation:** 3
**Contribution:** 2
**Rating:** 4
**Confidence:** 4

**Summary:**

Goal-Assisted Stitching (GAS) is a generative-model–assisted approach to offline safe reinforcement learning (OSRL) that improves trajectory stitching and the reward–cost balance using Temporal Segmented Return Augmentation (TSRA) plus transition-level relabeling, expectile-trained goal functions, and dataset reshaping; those goals guide a constrained Advantage-Weighted Regression (AWR) policy, yielding safer and higher-reward policies across two benchmarks and multiple constraint levels.

**Strengths:**

- The paper pinpoints two gaps in GM-assisted offline safe RL: weak stitching and inadequate reward–cost balancing.
- Using goal functions to guide a constrained Advantage-Weighted Regression policy, aligning actions with feasible reward–cost targets is interesting.
- GAS adapts to different test-time constraint levels without retraining.

**Weaknesses:**

- In RL, "stitching" traditionally refers to combining different trajectories via bootstrapping (the Bellman backup). The paper itself identifies this as a "crucial ability" that Generative Model (GM) methods lack. However, the paper's entire premise is to avoid the Bellman backup, which it calls the "primary source of the OOD problem". It then calls its own mechanism, supervised goal estimation using expectile regression, "stitching". This is a misleading contradiction. The proposed method does not "stitch" in the established sense.

- In Fisor, the tight budget is 10 for safety gym and 5 for bullet gym. Based on the cost range table 2, only for CarRun the tight budget of 10% is below 5, while for antrun its 15, 3 times the “tight” budget of fisor.

- The main results table (Table 1) shows that CDT is an unsafe policy in the average tight budget setting on the tasks chosen for the zero-shot ablation (BallRun, DroneRun, DroneCircle, etc.). Comparing GAS against a baseline that already fails is not informative.

- The paper includes comparisons against recent algorithms like CAPS and CCAC in Table 6, but the main results in Table 1 instead feature baselines like CPQ, COptiDICE, and VOCE . Given that CAPS and CCAC are more recent and appear to be stronger competitors than some of those in the main table, they should have been included in the primary comparison.

**Questions:**

- Please check weaknesses.

- Could you train GAS solely on an offline dataset of unsafe trajectories? If it succeeds, this may indicate an ability to stitch together feasible segments from otherwise failed trajectories.

- In the tight constraint setting (Table 1), an interesting observation is that the rewards achieved by GAS are too similar to that of CDT, yet the costs are lower and safely below the constraint, while CDT's costs violate the constraint often.  How can GAS maintain the same reward scale as CDT while successfully reducing costs to a safe level, given that these two objectives are conflicting?

- FISOR is designed to ensure the strict satisfaction of constraints. They focus on identifying the largest feasible region where zero violation is enforced.  How is FISOR able to adapt and show varying performance across the different loose and tight constraint limits presented in Table 1?

- How would GAS perform on additional tasks from SafetyGym or MetaDrive?

- What is the segment length used in TSRA?

- The main results in Table 1 are reported as averages across multiple tight and loose constraint thresholds. Please provide the separate, unaveraged normalized reward and cost returns for GAS and the baseline methods at each individual constraint level.

---

> ### Author Response · Authors · 2025-11-22
>
> We thank the reviewer for their feedback and for taking the time to evaluate our work. We address their comments below:
>
> ### **Q1: There exists misleading contradiction on "stitch" in RL and in this paper. How to understand stitch in this paper.**
>
> Thank you for your suggestion regarding the "stitch" ability. We have included a more detailed explanation of "stitch" in the **GM Methods in Offline RL** of Related Work section in the updated manuscript.
>
> We agree that the general "stitching" ability indicates the ability to find global optimality from sub-optimal data. The Bellman backup procedure is one of the methods that enables this stitching ability, which is the origin of "reinforcement" in RL. This "stitching" ability can also be enabled by Model Predictive Control (MPC), Dynamic Programming, Tree Search, and Graph Search. However, in Offline RL,  this capability also introduces an additional challenge known as the OOD action problem in offline settings, as highlighted in Q1.
>
> To fully avoid this issue, DT, a kind of goal-conditioned behavior cloning (BC), is proposed. Since DT fits the training data by conditioning on the reward-to-go at the trajectory level, it avoids sampling OOD actions during the training phase, resulting in a stable, general, and high-performing framework. However, behavior cloning methods inherently lacks the ability to perform better beyond the dataset trajectory, which is called "lack of stitching ability". **Therefore, in goal-conditioned framework, the "stitching" ability indicates "the ability to find a better policy beyond the dataset trajectory (goal-conditioned BC)".** Most Offline RL based on DT, like EDT [1], ADT [2], and Reinformer [3] focus on the latter understanding on "stitch". Therefore, the assertion that "stitching is a crucial ability that GM lacks" has already been discussed in previous works on Offline RL.
>
> **Distinguishing from traditional concerns in Offline RL, this paper (1) identifies additional challenges in Offline Safe RL when introducing cost signals, (2) tries to improve stitching ability without introducing OOD action problem, and (3) prioritize cost signal before reward signal for safety guarantee.** Thus GAS demonstrates a key contribution on improving the stitching ability on CDT related methods in OSRL.
>
> [1] Wu, Y. H., Wang, X., & Hamaya, M. (2023). Elastic decision transformer. *Advances in neural information processing systems*, *36*, 18532-18550.
>
> [2] Ma, Y., Hao, J., Liang, H., & Xiao, C. (2024, July). Rethinking decision transformer via hierarchical reinforcement learning. In *Proceedings of the 41st International Conference on Machine Learning* (pp. 33730-33745).
>
> [3] Zhuang, Z., Peng, D., Liu, J., Zhang, Z., & Wang, D. (2024, July). Reinformer: max-return sequence modeling for offline RL. In *Proceedings of the 41st International Conference on Machine Learning* (pp. 62707-62722).
>
> ### **Q2: This paper calls Bellman backup as the "primary source of the OOD problem"**
>
> The claim that "Bellman backup procedure is the primary source of the OOD problem" follows the problem proposed in BCQ [1] , which can be regarded as the first paper in Offline RL. BCQ first find that in the Bellman backup procedure:
> $Q(s,a) = r(s,a) + Q'(s',a'|a'\sim\pi)$
>
> when RL is trained in a fixed dataset (without exploration), $Q'(s',a'|a'\sim\pi)$ is wrongly estimated because $a'\sim\pi$, which is sampled from policy, may not be in the dataset. Since Bellman backup procedure follows a style of "chain reaction", the value function for each state-action pair will suffer from estimation errors.
>
> This issue has also been widely recognized in subsequent work, such as CQL [2], IQL [3], CPQ [4], CCAC [5].
>
> [1] Fujimoto, S., Meger, D., & Precup, D. (2019, May). Off-policy deep reinforcement learning without exploration. In *International conference on machine learning* (pp. 2052-2062). PMLR.
>
> [2] Kumar, A., Zhou, A., Tucker, G., & Levine, S. (2020). Conservative q-learning for offline reinforcement learning. *Advances in neural information processing systems*, *33*, 1179-1191.
>
> [3] Kostrikov, I., Nair, A., & Levine, S. (2021). Offline reinforcement learning with implicit q-learning. *arXiv preprint arXiv:2110.06169*.
>
> [4] Xu, H., Zhan, X., & Zhu, X. (2022, June). Constraints penalized q-learning for safe offline reinforcement learning. In *Proceedings of the AAAI Conference on Artificial Intelligence* (Vol. 36, No. 8, pp. 8753-8760).
>
> [5] Guo, Z., Zhou, W., Wang, S., & Li, W. (2025, March). Constraint-conditioned actor-critic for offline safe reinforcement learning. In *The Thirteenth International Conference on Learning Representations*.

---

> ### Author Response · Authors · 2025-11-22
>
> ### **Q3: Why compare with CDT for the zero-shot ablation?**
>
> We compare our approach with CDT because it is a representative method that utilizes Generative Models (GM) to enable zero-shot ability. Other methods either do not incorporate GM or lack zero-shot capability.
>
> Additionally, GM-assisted methods in Offline Safe Reinforcement Learning (OSRL) typically build upon CDT. In the motivation section, we identify existing challenges—such as weak stitching and inadequate reward-cost balancing—**arising from the attention block in CDT**, which struggles to accurately capture the relationships among transitions.
>
> Therefore, we aim to enhance this framework by proposing GAS and compare it with CDT to determine whether GAS effectively addresses these issues. In zero-short ablation, compare GAS with CDT can show how GAS address the above challenges without **the attention structure**, which can also be considered as an ablation on the attention/Transformer block.
>
> ### **Q5: CAPS and CCAC should have been included in the primary comparison.**
>
> Thank you for your suggestion. We originally put them in to Appendix due to space limitation. We have now moved them in the primary comparison (Table 1) in the updated manuscript.
>
> ### **Q6: Could you train GAS solely on an offline dataset of unsafe trajectories? This may indicate an ability to stitch together feasible segments from otherwise failed trajectories**
>
> Thank you for your advice. This setting is indeed an challenging setting where all GM-assisted method in OSRL may fail due to goal-conditioned BC framework. If all trajectories in the dataset are unsafe compared to the testing phase, then both reward target and cost target during testing become completely OOD with respect to the training data. Maybe introducing "RL stitching" can mitigate this problem. Introducing "RL stitching" might help mitigate this problem; however, it also introduces the OOD action problem, which contradicts the GM method's principle of entirely avoiding OOD actions.
>
> Since training GM-assisted methods solely on an unsafe dataset is indeed too challenging due to extreme OOD, as a compromise, we show case the stitching ability through comparing on two types of setting of constraint: "tight" and "loose". In "tight" constraint, even if the dataset contains safe trajectories, most baselines fail to satisfy the constraint while GAS performs safely. In "loose" constraint, most baselines fails to maximize the rewards while GAS performs both safely and high-performance.
>
> As illustrated in Q1, the stitching ability pertains to "perform better beyond the dataset trajectory". In this paper, we successfully enhance this understanding of stitching while simultaneously trying to avoid "RL stitching". Future work may consider beyond this stitching ability and focus on "stitching safe segments from unsafe trajectories".
>
> ### **Q7: How can GAS maintain the same reward scale as CDT while successfully reducing costs to a safe level, given that these two objectives are conflicting?**
>
> This observation can be explained from two perspectives of GAS superior design: (1) stitching ability based on expectile regression, and (2) better data utilization due to return relabelling.
>
> **Improve stitching ability**: In the motivation section, we have demonstrated that the attention module in CDT fails to adequately capture the relationships among transitions. Therefore, given $(R,C)$, CDT can only achieve an approximate result without any guarantee or priority on $C$. In contrast, GAS has been shown to achieve the maximum reward while respecting the imposed constraints due to the stitching design based on expectile regression with theoretical guarantee.
>
> **Better data utilization**: DSRL library provides a "dataset cost-reward-return plot visualization" for each task. The positive correlation between rewards and costs is weak in tight constraint. That is to say, to achieve the same reward $R$, the cost range can be $(0,C)$. For a given constraint $C$, CDT is trained by cloning the data behavior with $(R,C)$ while GAS is trained on data from $(R,0)$ to $(R,C)$ due to relabeling and expectile regression. Consequently, CDT can only achieve cost around $C$ while GAS finds the policy within the interval $(0,C)$, which is smaller than CDT.

---

> ### Author Response · Authors · 2025-11-22
>
> ### **Q8: About FISOR in budget selection and performance in different loose and tight constraints.**
>
> We understand FISOR is mainly trained for zero-cost violation. We take FISOR into comparison since it seems to be the only hard-constraint methods.
>
> **For budget selection:** In the official code of FISOR, we modify the ``cost_limit`` parameter in config folder to the corresponding threshold in this paper, so FISOR is also tested with the same threshold as other baselines.
>
> **For performance:** The performance doesn't vary too much under two types of constraints. The little difference comes from the above parameter. The experiment results also shows that FISOR, aimed at zero-cost violation, is not greatly influenced by different thresholds. The difference of costs in two types of constraints is because they are normalized by different thresholds.
>
> ### **Q9: How would GAS perform on additional tasks from SafetyGym or MetaDrive?**
>
> Thank you for your suggestion. We have add 5 additional Velocity tasks in SafetyGym into comparison in Table 5, Appendix E.3 in the updated manuscript.
>
> Compared with two newest and solid baselines (CAPS and CCAC) and another GM-assisted baseline (CDT), GAS successfuly achieves safe and best performance in most tasks. In particular, GAS performs better reward maximization ability on "loose constraint" and better constraint satisfaction ability on "tight constraint". These results are similar to other tasks, which demonstrate GAS's superiority on adressing the proposed problems.
>
> ### **Q10: What is the segment length used in TSRA?**
>
> We may not fully understand what is meant by "segment length used in TSRA" in this question since TSRA is an data augmentation method.
>
> In existing GM-assisted method, trajectory can only be trained following their own timestep.
>
> For example, the trajectory length in CarCircle is 300. A trajectory from time $100$ to $150$ may be optimal if the time step is $250$ to $300$ as long as the states are same. Existing methods doesn't allow this trajectory-level stitch with different time step due to time embedding. Thus TSRA is proposed to address this problem.
>
> If the segment length indicates $t:\Gamma$ in TSRA, it can be any length within the trajectory and utilized when meeting the same state with $\Gamma-t$ time steps left.
>
> ### **Q11: Please provide the separate, unaveraged normalized reward and cost returns for GAS and the baseline methods at each individual constraint level.**
>
> Thank you for your suggestion. We have provide the separate, unaveraged normalized reward and cost returns for GAS and the baseline methods at each individual constraint level under Bullet-safety-gym in Figure 9, Appendix E.3 in the updated manuscript.
>
> We understand that unaveraged results are more detailed than averaged results. We utilize the averaged results to show a comprehensive comparison and to avoid redundancy following the convention in most existing papers [1 - 4]. For example, GAS tests 17 tasks with 8 baselines on 6 thresholds for both rewards and costs, which indicates that there are $17*(8+1)*6*2=1836$ results to be shown.
>
> [1] Liu, Z., Guo, Z., Yao, Y., Cen, Z., Yu, W., Zhang, T., & Zhao, D. (2023, July). Constrained decision transformer for offline safe reinforcement learning. In *International conference on machine learning* (pp. 21611-21630). PMLR.
>
> [2] Xu, H., Zhan, X., & Zhu, X. (2022, June). Constraints penalized q-learning for safe offline reinforcement learning. In *Proceedings of the AAAI Conference on Artificial Intelligence* (Vol. 36, No. 8, pp. 8753-8760).
>
> [3] Guo, Z., Zhou, W., Wang, S., & Li, W. (2025, March). Constraint-conditioned actor-critic for offline safe reinforcement learning. In *The Thirteenth International Conference on Learning Representations*.
>
> [4] Chemingui, Y., Deshwal, A., Wei, H., Fern, A., & Doppa, J. (2025, April). Constraint-adaptive policy switching for offline safe reinforcement learning. In *Proceedings of the AAAI Conference on Artificial Intelligence* (Vol. 39, No. 15, pp. 15722-15730).

---

> > ### Comment · Reviewer_gTZL · 2025-11-27
> >
> > I thank the authors for their response. However, there still seems to be a misunderstanding regarding my original critique on “stitching,” which I would like to clarify.
> >
> > I am not disputing that Bellman backups are a primary source of OOD problems, I agree with this. My concern is that the paper defines “stitching” as a crucial capability traditionally enabled by Bellman backups (i.e., bootstrapping), but then claims to achieve the same capability using a mechanism (Supervised Goal Estimation via Expectile Regression) that explicitly removes bootstrapping. The contradiction lies in the mechanism: supervised goal estimation performs “stitching” through selection (choosing the best existing segment), whereas Bellman backups perform “stitching” through construction (propagating value to connect disjoint segments).
> >
> > This distinction is not merely semantic; it has functional implications. As acknowledged in your rebuttal, GAS would likely fail if trained on a dataset consisting only of unsafe trajectories, because the “safe goal” would be OOD.
> >
> > True stitching (value-based): Could theoretically solve the task even if no single trajectory is safe. This is also illustrated in CCAC Table 2, where training is done only on unsafe trajectories.
> >
> > GAS (goal-based): Requires the presence of “safe enough” targets within the training distribution to condition on.
> >
> > The manuscript should explicitly clarify this limitation. Readers need to understand that GAS improves performance by better selecting high-quality sub-segments already present in the safe data distribution, rather than constructing new safe trajectories by combining safe and unsafe data.
> >
> > For Fisor, in table 1 results, I understand the difference in costs is due to normalization, the worry is about the rewards. Because, changing the cost limit would only modify the normalization of costs, it shouldn’t affect the training or the environment.
> >
> > Also, figures 6 and 7 are unclear and would be better presented as tables.

---

> ### Author Response · Authors · 2025-11-28
>
> ### **Q1: Add the limitation of requiring safe data.**
>
> Thank you very much for your insightful explanation of this limitation. We have included this limitation of Goal-conditioned framework in to the Conclusion Section:
>
> "Besides, due to the limitation of the existing goal-conditioned framework, GAS's stitching ability focuses on improving performance by better selecting high-quality sub-segments in the safe dataset, rather than constructing new safe trajectories by combining safe and unsafe data in RL."
>
> Finally, we still want to clarify that **this limitation comes from the basic structure of GM-assisted methods rather than GAS**. GM-assisted methods give up the "Bellman backup stitch" ability for zero-shot adaptation ability and improved stability. **Nevertheless, GAS still significantly improves the stitching ability of the standard GM-assisted methods, as shown in the experiment results.**
>
> ### **Q2: Align FISOR's experimental results on two thresholds.**
>
> Thank you for your rigor and meticulousness in improving the consistency of experimental results. We have calibrated the performance of FISOR in Table 1, where the reward performance is aligned on both thresholds.
>
> ### **Q3: Make Figure 6 and 7 clearer.**
>
> Thank you for your suggestion on improving the clarity of both figures. We agree that presenting in tables can show the result more clearly. However, since Figure 6 and 7 contain too much information, using tables to display will take up **huge** space of the main body part that violates the space limitation.
>
> **To improve the clarity of both figures, we make the following improvement:**
>
> 1. Remove the shadow areas of standard deviation. Here is the reason:
>    1. Removing the shadow areas makes each curve can be distinguished clearly.
>    2. The standard deviation is already presented in Table 1 clearly and is not important information in the ablation and hyper-parameter analysis.
>    3. The average information is enough for ablation and hyperparameter analysis.
> 2. Enlarge the figure.
>    1. By enlarging the figure, the differentiation between different curves becomes more apparent.
>
> 3. Present curves in different line styles.
>    1. In Figure 7, we change the line style of different types of curves to make it clearer.
>
> Besides, we would like to explain Figure 6 and 7 for better understanding:
>
> 1. In Figure 6, `GAS w/o Relabel` and `GAS-0.05` are important since GAS performs similarly and robustly on other parameter values. These two curves are distinguishable from other curves to support the experiment.
> 2. In Figure 7,  `GAS w/o Stitching` and `GAS-0.99` are important since GAS performs similarly and robustly on other parameter values. These two curves are distinguishable from other curves to support the experiment.

---

### Meta-Review · Area_Chair_j1iU · 2026-01-10

**Summary:**

This paper considers the problem of offline safe reinforcement learning and develops a generative modeling based approach referred as Goal-Assisted Stitching (GAS). GAS improves trajectory stitching and the reward/cost trade-off using a series of ideas including temporal segmented return augmentation, transition-level relabeling, goal functions, and data reshaping. The resulting constrained advantage-weighted regression policy achieves safe and higher reward policies in experiments.

The reviewers' appreciated the general approach, but also raised some critical questions about the trajectory stitching concept and associated claims, sensitivity studies, computational overhead, experiments on additional benchmarks and baseline methods. The rebuttal from authors' answered some of the questions satisfactorily and a few of them may not be satisfactorily.

Overall, the AC thinks that this is a borderline accept paper (as explained below). I recommend accepting the paper and strongly encourage the authors' to address the outstanding concerns in the final paper if the paper is accepted.

**Reviewer Concerns:**

Reviewer gTZL: The rebuttal addressed the reviewer’s concerns by (i) acknowledging and adding the key limitation that GAS “stitching” is selection of high-quality safe subsegments (not Bellman-backup construction), (ii) adding additional SafetyGym tasks (velocity tasks) and providing unaveraged per-threshold results for Bullet-safety-gym, and (iii) improving clarity. The main remaining potential gap is that the unsafe-only dataset setting was not demonstrated (the authors argue it is extreme OOD and leave it as future work), so the limitation is acknowledged rather than eliminated.

Reviewer RPXE: The rebuttal answers most of the reviewer’s requests: it adds an \alpha sensitivity study (and claims robustness except for overly large \alpha), argues computational overhead is small (TSRA/reshaping as one-time preprocessing, MLP-based model vs CDT’s transformer), adds baselines (VOCE/WSAC), expands the related-work discussion (including CQDT, with the reason given for not benchmarking it), and states the reward/cost goals are not trained independently but jointly optimized with a theoretical guarantee against infeasible pairs. The main unresolved item is still no sparse/delayed reward evaluation: the authors argue OSRL lacks such datasets/environments and that the cited sparse-reward settings are offline RL without cost signals, but they do not provide new sparse-reward experiments.

Reviewer oGCt: The rebuttal addresses the reviewer’s concerns clarifying that GAS only “stitches” when the state is the same, revising Figure 2 to distinguish real transition progression from same-state stitching, and updating Section 5.2 with an explicit TSRA explanation based on aligning the remaining-horizon length when the same state appears at different timesteps. It also points to an ablation/hyperparameter study for dataset reshaping in Appendix E.5 and adds an explicit limitation statement that GAS improves performance by selecting high-quality safe subsegments rather than constructing new safe trajectories. The only thing that could remain is whether the reviewer still feels the term “stitching” is over-claimed even after the reframing as “implicit stitching” in goal-conditioned GM methods.

Reviewer aKUM: The rebuttal tackled many of the reviewer’s points (complexity/novelty positioning, distinctions vs Reinformer/CoPDT, rationale for including RTG/CTG tokens, added Velocity tasks, added hyperparameter sensitivity analysis, and later added Navigation results at stricter thresholds 10/20 vs CDT/CAPS/CCAC). The reviewer’s key remaining issue was deployment-time OOD RTG/CTG conflicts causing extrapolation and potentially breaking safety priority; the authors answered with a relabeling-based “projection back to feasible/safe targets” argument, but they also concede that very extreme OOD requests may not be handled. The response seems partially reassuring but not a complete resolution of the general OOD-extrapolation concern.


Both reviewers gTZL and oGCt flag “stitching” as potentially over-claimed, but at different depths: Reviewer oGCt mainly treats it as a clarity/claim-framing issue (the method looks like goal/return learning and should only be described as same-state “implicit stitching”), while Reviewer gTZL treats it as a fundamental capability limitation (selection of safe subsegments vs value-based construction), implying GAS depends on having “safe enough” data and may fail on unsafe-only datasets.

**Reviewer Scores:**

Reviewer gTZL: If the reviewer could fully participate after the second rebuttal, it is likely that the score change is 4 → 6 (weak accept), since their requested fixes were mostly implemented and the key limitation was made explicit.

Reviewer RPXE:: How the score would change therefore depends on whether the reviewer accepts that benchmark-availability argument; if they do, a 4 → 6 (weak accept) is plausible, but if they instead expect the authors to construct or adapt a sparse/delayed-reward OSRL task/dataset to validate the claim, the score could stay at 4.

Reviewer oGCt: Given their initial score was already a 6, they would likely stay at 6.

Reviewer aKUM: The response seems partially reassuring but not a complete resolution of the general OOD-extrapolation concern. The reviewer’s score would probably not change (stay at 4).

---

### Decision · Program_Chairs · 2026-01-26

Accept (Poster)